# Adversarial Causal Augmentation for Graph Covariate Shift

## Abstract

Out-of-distribution (OOD) generalization on graphs is drawing widespread attention. However, existing efforts mainly focus on the OOD issue of correlation shift. While another type, covariate shift, remains largely unexplored but is the focus of this work. From a data generation view, causal features are stable substructures in data, which play key roles in OOD generalization. While their complementary parts, environments, are unstable features that often lead to various distribution shifts. Correlation shift establishes spurious statistical correlations between environments and labels. In contrast, covariate shift means that there exist unseen environmental features in test data. Existing strategies of graph invariant learning and data augmentation suffer from limited environments or unstable causal features, which greatly limits their generalization ability on covariate shift. In view of that, we propose a novel graph augmentation strategy: Adversarial Causal Augmentation (**AdvCA**), to alleviate the covariate shift. Specifically, it adversarially augments the data to explore diverse distributions of the environments. Meanwhile, it keeps the causal features stable across diverse environments. It maintains the environmental diversity while ensuring the invariance of the causal features, thereby effectively alleviating the covariate shift. Extensive experimental results with in-depth analyses demonstrate that AdvCA can outperform 14 baselines on synthetic and real-world datasets with various covariate shifts.

## 1 Introduction

Graph learning mostly follows the assumption that training and test data are independently drawn from an identical distribution. Such an assumption is difficult to be satisfied in the wild, due to out-of-distribution (OOD) issues (Shen et al., 2021), where the training and test data are from different distributions. Hence, OOD generalization on graphs is attracting widespread attention (Li et al., 2022b). However, existing studies mostly focus on correlation shift, which is just one type of OOD issue (Ye et al., 2022; Wiles et al., 2022). While another type, covariate shift, remains largely unexplored but is the focus of our work.

Covariate shift is in stark contrast to correlation shift *w.r.t.* causal and environmental features of data[1]. Specifically, from a data generation view, **causal features**[2] are the substructures of the entire graphs that truly reflect the predictive property of data, while their complementary parts are the **environmental features** that are noncausal to the predictions. Following prior studies (Arjovsky et al., 2019; Wu et al., 2022b), we assume causal features are stable across distributions, in contrast to the environmental features. Correlation shift denotes that environments and labels establish inconsistent statistical correlations in training and test data; whereas, covariate shift

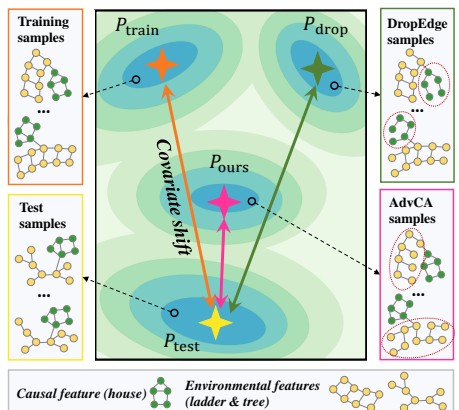

Figure 1: $P_{\text{train}}$ and $P_{\text{test}}$ denote the training and test distributions. $P_{\text{drop}}$ and $P_{\text{ours}}$ represent the distributions of augmented data via DropEdge and AdvCA. AdvCA establishes a smaller covariate shift ($\leftrightarrow$) with test distribution than DropEdge ($\leftrightarrow$).

---

[1] We provide detailed discussions of these two distribution shifts in Appendix C.

[2] We provide a formal definition in Assumption 1.

means that the environmental features in test data are unseen in training data (Ye et al., 2022; Wiles et al., 2022; Gui et al., 2022). For example, in Figure 1, the environmental features *ladder* and *tree* are different in training and test data, which forms the covariate shift (↔). Taking molecular property predictions as another example, functional groups (*e.g.,* nitrogen dioxide ($NO_2$)) are causal features that determine the predictive property of molecules. While scaffolds (*e.g.,* carbon rings) are irrelevant patterns (Wu et al., 2018), which can be seen as the environments. In practice, we often need to use molecular graphs collected in the past to train models, hoping that the models can predict the properties of molecules with new scaffolds in the future (Hu et al., 2020).

Because of the differences between correlation and covariate shifts, we take a close look at the existing efforts on graph generalization. Existing efforts (Li et al., 2022b) mainly fall into the following research lines, each of which has inherent limitations to solve covariate shift.

- **Invariant graph learning** (Wu et al., 2022b; Liu et al., 2022; Sui et al., 2022) gradually becomes a prevalent paradigm for OOD generalization. The main idea is to capture the causal features by minimizing the empirical risks within different environments. Unfortunately, it implicitly makes a prior assumption that all test environments are available during training. This assumption is unrealistic owing to the obstacle of training data covering all possible test environments. Learning in limited environments can only alleviate the spurious correlations that are hidden in the training data, but fail to extrapolate test distributions with unseen environments.
- **Graph data augmentation** (Ding et al., 2022; Zhao et al., 2022) perturbs graph features to enrich the distribution seen during training for better generalization. It can be roughly divided into node-level (Kong et al., 2022), edge-level (Rong et al., 2020), and graph-level (Wang et al., 2021; Han et al., 2022) with random (You et al., 2020) or adversarial strategies (Suresh et al., 2021). However, they are prone to destroy the causal features, which easily loses control of the perturbed distributions. For example, in Figure 1, the random strategy of DropEdge (Rong et al., 2020) will inevitably perturb the causal features (highlighted by red circles). As such, it fails to alleviate the covariate shift (↔), even degenerating the generalization ability.

Scrutinizing the limitations of the aforementioned studies, insufficient environments and unstable causal features largely hinder the ability of these generalization efforts against the covariate shift. Hence, we naturally ask a question: "*Can the augmented samples simultaneously preserve the diversity of environmental features and the invariance of causal features?*"

Towards this end, we first propose two principles for graph augmentation: environmental diversity and causal invariance. Specifically, environmental diversity encourages the augmentation to extrapolate unseen environments; meanwhile, causal invariance shortens the distribution gap between the augmented data and test data. To achieve these principles, we design a novel graph augmentation strategy: Adversarial Causal Augmentation (**AdvCA**). Specifically, we augment the graphs by a network, named adversarial augmenter. It adversarially generates the masks on edges and node features, which makes OOD exploration for improving the environmental diversity. To maintain the stability of the causal features, we adopt another network, named causal generator. It generates the masks that capture causal features. Finally, we delicately combine these masks and apply them to graph data. As shown in Figure 1, AdvCA only perturbs the environmental features, while keeping the causal parts untorched. Our quantitative experiments also verify that AdvCA can narrow the distribution gap between the augmented data and test data, as illustrated in Figure 1 (↔), thereby effectively overcoming the covariate shift issues. Our contributions can be summarized as:

- **Problem**: We are exploring one specific type of OOD issue in graph learning: covariate shift, which is of great need but largely unexplored.
- **Method**: We design a graph augmentation method, AdvCA, which focuses on covariate shift issues. It maintains the stability of causal features while ensuring environmental diversity.
- **Experiment**: We conduct extensive experiments on synthetic and real datasets. The experimental results with in-depth analyses demonstrate the effectiveness of AdvCA.

## 2 PRELIMINARIES

In this section, we first give the formal definitions of causal features, environmental features, and graph covariate shift. Then we present the problem of graph classification under covariate shift.

## 2.1 NOTATIONS

We define the uppercase letters (*e.g., G*) as random variables. The lower-case letters (*e.g., g*) are samples of variables, and the blackboard bold typefaces (*e.g., $\mathbb{G}$*) denote the sample spaces. Let $g = (\mathbf{A}, \mathbf{X}) \in \mathbb{G}$ denote a graph, where $\mathbf{A}$ and $\mathbf{X}$ are its adjacency matrix and node features, respectively. It is assigned with a label $y \in \mathbb{Y}$ with a fixed labeling rule $\mathbb{G} \to \mathbb{Y}$. Let $\mathcal{D} = \{(g_i, y_i)\}$ denote a dataset that is divided into a training set $\mathcal{D}_{\mathrm{tr}} = \{(g_i^e, y_i^e)\}_{e \in \mathcal{E}_{\mathrm{tr}}}$ and a test set $\mathcal{D}_{\mathrm{te}} = \{(g_i^e, y_i^e)\}_{e \in \mathcal{E}_{\mathrm{te}}}$. $\mathcal{E}_{\mathrm{tr}}$ and $\mathcal{E}_{\mathrm{te}}$ are the index sets of training and test environments, respectively. In this work, we focus on graph classification scenario, which aims to train models with $\mathcal{D}_{\mathrm{tr}}$ and infer the labels in $\mathcal{D}_{\mathrm{te}}$.

## 2.2 DEFINITIONS AND PROBLEM FORMATIONS

Following studies (Arjovsky et al., 2019; Wu et al., 2022b), we assume that the inner mechanism of the labeling rule $\mathbb{G} \to \mathbb{Y}$ usually depends on the causal features, which are particular subparts of the entire data. Causal invariance denotes that the relationship between the causal feature and label is invariant across different environments or distributions, which makes OOD generalization possible (Ye et al., 2022). While the complement of causal parts, environmental features, are noncausal for predicting the graphs. Now we give a formal definition of these features.

**Assumption 1 (Causal & Environmental Feature)** *Assume input graph $G$ containing two features $G_{\mathrm{cau}}$, $G_{\mathrm{env}}$, and they satisfy: $G_{\mathrm{cau}} \cup G_{\mathrm{env}} = G$. If they obey the following conditions: i) (sufficiency condition) $P(Y|G_{\mathrm{cau}}) = P(Y|G)$; ii) (independence condition) $Y \perp\!\!\!\perp G_{\mathrm{env}} \mid G_{\mathrm{cau}}$, then we define $G_{\mathrm{cau}}$ and $G_{\mathrm{env}}$ as the causal feature and environmental feature, respectively.*

Sufficiency condition requires that causal features should be sufficient to preserve the critical information of data $G$ related to the label $Y$. While the independence indicates that causal feature can shield the label from the influence of the environment feature. It makes causal features establish an invariant relationship with labels across different environments. Hence, distribution shifts are only caused by the environmental features rather than causal features. Recent studies (Ye et al., 2022; Gui et al., 2022) have pointed out that OOD issue can be specifically divided into correlation shift and covariate shift. Since we mainly focus on the latter, we put detailed discussions between them in Appendix C. Now we give a formal definition of the covariate shift on graphs.

**Definition 1 (Graph Covariate Shift)** *Let $P_{\mathrm{tr}}$ and $P_{\mathrm{te}}$ denote the probability functions of the training and test distributions. We measure the covariate shift between distributions $P_{\mathrm{tr}}$ and $P_{\mathrm{te}}$ as*

$$\mathrm{GCS}(P_{\mathrm{tr}}, P_{\mathrm{te}}) = \frac{1}{2} \int_{\mathcal{S}} |P_{\mathrm{tr}}(g) - P_{\mathrm{te}}(g)| dg, \tag{1}$$

*where $\mathcal{S} = \{g \in \mathbb{G} | P_{\mathrm{tr}}(g) \cdot P_{\mathrm{te}}(g) = 0\}$, which covers the features (e.g., environmental features) that do not overlap between the two distributions.*

$\mathrm{GCS}(P_{\mathrm{tr}}, P_{\mathrm{te}})$ is always bounded in $[0, 1]$. The issue of graph covariate shift is very common in practice. For example, the chemical properties of molecules are mainly determined by specific functional groups, which can be regarded as causal features to predict these properties (Arjovsky et al., 2019; Wu et al., 2022b). While their scaffold structures (Wu et al., 2018), which are often irrelevant to their properties, can be seen as environmental features. In practice, we often need to train models on past molecular graphs, and hope that the model can predict the properties of future molecules with novel scaffolds (Hu et al., 2020). Hence, this work focuses on the covariate shift issues. Now we give a formal definition of this problem as follows.

**Problem 1 (Graph Classification under Covariate Shift)** *Given the training and test sets with environment sets $\mathcal{E}_{\mathrm{tr}}$ and $\mathcal{E}_{\mathrm{te}}$, they follow distributions $P_{\mathrm{tr}}$ and $P_{\mathrm{te}}$, and they satisfy: $\mathrm{GCS}(P_{\mathrm{tr}}, P_{\mathrm{te}}) > 0$. We aim to use the data collected from training environments $\mathcal{E}_{\mathrm{tr}}$, and learn a powerful graph classifier $f^* : \mathbb{G} \to \mathbb{Y}$ that performs well in all possible test environments $\mathcal{E}_{\mathrm{te}}$:*

$$f^* = \arg\min_{f} \sup_{e \in \mathcal{E}_{\mathrm{te}}} \mathbb{E}^e [\ell(f(g), y)], \tag{2}$$

*where $\mathbb{E}^e [\ell(f(g), y)]$ is the empirical risk on the environment $e$, and $\ell(\cdot, \cdot)$ is the loss function.*

Problem 1 states that it is unrealistic for the training set to cover all possible environments in the test set. It means that we have to extrapolate unknown environments by using the limited training environments at hand, which makes this problem more challenging.

## 3 METHODOLOGY

In this section, we first propose two principles for graph data augmentation. Guided by these principles, we design a new graph augmentation strategy that can effectively solve Problem 1.

### 3.1 TWO PRINCIPLES FOR GRAPH AUGMENTATION

Scrutinizing Problem 1, we observe that the covariate shift is mainly caused by the scarcity of training environments. Existing efforts (Liu et al., 2022; Sui et al., 2022; Wu et al., 2022b) make intervention or replacement of the environments to capture causal features. However, these environmental features still stem from the training distribution, which may result in a limited diversity of the environments. Worse still, if the environments are too scarce, the model will inevitably learn the shortcuts between these environmental features, resulting in suboptimal learning of the causal parts. To this end, we propose the first principle for data augmentation:

**Principle 1 (Environmental Diversity)** *Given a set of graphs $\{g\}$ with distribution function $P$. Let $T(\cdot)$ denote an augmentation function that augments graphs $\{T(g)\}$ to distribution function $\widetilde{P}$. Then $T(\cdot)$ should meet $\mathrm{GCS}(P, \widetilde{P}) \to 1$.*

Principle 1 states that $\widetilde{P}$ should keep away from the original distribution $P$. Hence, the distribution of augmented data tends not to overlap with the original distribution, which encourages the diversity of environmental features. However, from a data generation perspective, causal features are stable and shared across environments (Kaddour et al., 2022), so they are essential features for OOD generalization. Since Principle 1 does not expose any constraint on the invariant property of the augmented distribution, we here propose the second principle for augmentation:

**Principle 2 (Causal Invariance)** *Given a set of graphs $\{g\}$ with a corresponding causal feature set $\{g_{\mathrm{cau}} = (\mathbf{A}_{\mathrm{cau}}, \mathbf{X}_{\mathrm{cau}})\}$. Let $T(\cdot)$ denote an augmentation function that augments graphs $\{T(g)\}$ with a corresponding causal feature set $\{\widetilde{g}_{\mathrm{cau}} = (\widetilde{\mathbf{A}}_{\mathrm{cau}}, \widetilde{\mathbf{X}}_{\mathrm{cau}})\}$. Then $T(\cdot)$ should meet $\mathbb{E}[\|\mathbf{A}_{\mathrm{cau}} - \widetilde{\mathbf{A}}_{\mathrm{cau}}\|_F^2] \to 0$ and $\mathbb{E}[\|\mathbf{X}_{\mathrm{cau}} - \widetilde{\mathbf{X}}_{\mathrm{cau}}\|_F^2] \to 0$, where $\|\cdot\|_F^2$ is the Frobenius norm.*

Principle 2 emphasizes the invariance of the graph structures and node features in causal parts after data augmentation. As illustrated in Figure 1, Principle 1 keeps the distribution of augmented data away from the training distribution; meanwhile, Principle 2 restricts the distribution of augmented data not too far from the test distribution. These two principles complement each other, and further cooperate together to alleviate the covariate shift.

### 3.2 OUT-OF-DISTRIBUTION EXPLORATION

Given a GNN model $f(\cdot)$ with parameters $\theta$, we decompose $f = \Phi \circ h$, where $h(\cdot) : \mathbb{G} \to \mathbb{R}^d$ is a graph encoder to yield $d$-dimensional representations, and $\Phi(\cdot) : \mathbb{R}^d \to \mathbb{Y}$ is a classifier. To comply with Principle 1, we need to do OOD exploration. Inspired by distributionally robust optimization (Sagawa et al., 2020), we consider the following optimization objective:

$$\min_{\theta} \left\{ \sup_{\widetilde{P}} \{ \mathbb{E}_{\widetilde{P}}[\ell(f(g), y)] : D(\widetilde{P}, P) \le \rho \} \right\}, \tag{3}$$

where $P$ and $\widetilde{P}$ denote the original and explored data distributions, respectively. $D(\cdot, \cdot)$ is a distance metric between two probability distributions. The solution to Equation 3 guarantees the generalization within a distance $\rho$ of the distribution $P$. To better measure the distance between distributions, as suggested by Sinha et al. (2018), we adopt the Wasserstein distance (Arjovsky et al., 2017; Volpi et al., 2018) as the distance metric. The distance metric function can be defined as:

$$D(\widetilde{P}, P) := \inf_{\mu \in \Gamma(\widetilde{P}, P)} \mathbb{E}_{\mu}[c(\widetilde{g}, g)], \tag{4}$$

where $\Gamma(\widetilde{P}, P)$ is the set of all couplings of $\widetilde{P}$ and $P$, $c(\cdot, \cdot)$ is the cost function. Studies (Dosovitskiy & Brox, 2016; Volpi et al., 2018) also suggest that the distances in representation space typically correspond to semantic distances. Hence, we define the cost function in the representation space and give the following transportation cost:

$$c(\widetilde{g}, g) = \|h(\widetilde{g}) - h(g)\|_2^2. \tag{5}$$

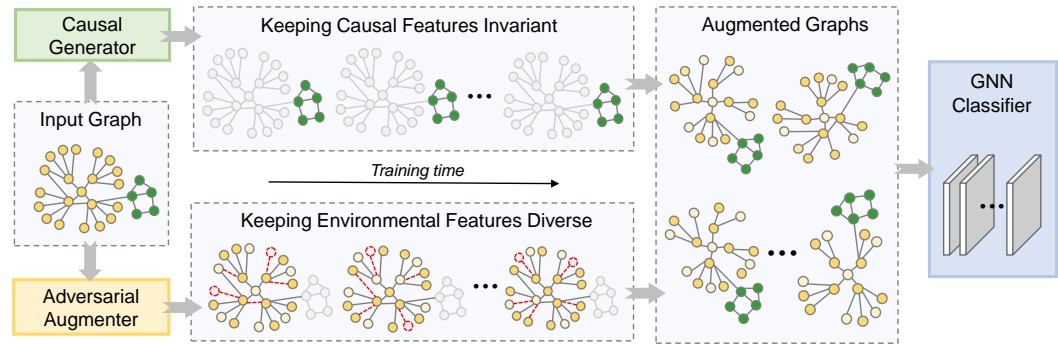

Figure 2: The overview of Adversarial Causal Augmentation (AdvCA) Framework.

It denotes the "cost" of augmenting the graph $g$ to $\widetilde{g}$. We can observe that it is difficult to set a proper $\rho$ in Equation 3. Instead, we consider the Lagrangian relaxation for a fixed penalty coefficient $\gamma$. Inspired by Sinha et al. (2018), we can reformulate Equation 3 as follows:

$$\min_{\theta} \left\{ \sup_{\widetilde{P}} \{ \mathbb{E}_{\widetilde{P}} [\ell(f(g), y)] - \gamma D(\widetilde{P}, P) \} = \mathbb{E}_P [\phi(f(g), y)] \right\}, \tag{6}$$

where $\phi(f(g), y) := \sup_{\widetilde{g} \in \mathbb{G}} \{ \ell(f(\widetilde{g}), y) - \gamma c(\widetilde{g}, g) \}$. And we define $\phi(f(g), y)$ as the robust surrogate loss. If we conduct gradient descent on the robust surrogate loss, we will have:

$$\nabla_{\theta} \phi(f(g), y) = \nabla_{\theta} \ell(f(\widetilde{g}^*), y), \tag{7}$$

$$\text{where } \widetilde{g}^* = \arg\max_{\widetilde{g} \in \mathbb{G}} \{ \ell(f(\widetilde{g}), y) - \gamma c(\widetilde{g}, g) \}. \tag{8}$$

$\widetilde{g}^*$ is an augmented view of the original data $g$. Hence, to achieve OOD exploration, we just need to perform data augmentation via Equation 8 on the original data $g$.

## 3.3 ADVERSARIAL CAUSAL AUGMENTATION

Equation 8 endows the ability of OOD exploration to data augmentation, which makes the augmented data meet Principle 1. In addition, to achieve Principle 2, we also need to implement causal feature learning based on the sufficiency and independence conditions in Assumption 1. Hence, we design a novel graph augmentation strategy: Adversarial Causal Augmentation (**AdvCA**). The overview of the proposed framework is depicted in Figure 2, which mainly consists of two components: adversarial augmenter and causal generator. Adversarial augmenter achieves OOD exploration through adversarial data augmentation, which encourages the diversity of environmental features; meanwhile, the causal generator keeps causal feature invariant by identifying causal features from data. Below we elaborate on the implementation details.

**Adversarial Augmenter & Causal Generator.** We design two networks, adversarial augmenter $T_{\theta_1}(\cdot)$ and causal generator $T_{\theta_2}(\cdot)$, which generate masks for nodes and edges of graphs. They have the same structure and are parameterized by $\theta_1$ and $\theta_2$, respectively. Given an input graph $g = (\mathbf{A}, \mathbf{X})$ with $n$ nodes, mask generation network first obtains the node representations via a GNN encoder $\widetilde{h}(\cdot)$. To judge the importance of nodes and edges, it adopts two MLP networks $\mathrm{MLP}_1(\cdot)$ and $\mathrm{MLP}_2(\cdot)$ to generate the soft node mask matrix $\mathbf{M}^x \in \mathbb{R}^{n \times 1}$ and edge mask matrix $\mathbf{M}^a \in \mathbb{R}^{n \times n}$ for graph data, respectively. In summary, the mask generation network can be decomposed as:

$$\mathbf{Z} = \widetilde{h}(g), \quad \mathbf{M}_i^x = \sigma(\mathrm{MLP}_1(\mathbf{h}_i)), \quad \mathbf{M}_{ij}^a = \sigma(\mathrm{MLP}_2([\mathbf{z}_i, \mathbf{z}_j])), \tag{9}$$

where $\mathbf{Z} \in \mathbb{R}^{n \times d}$ is node representation matrix, whose $i$-th row $\mathbf{z}_i = \mathbf{Z}[i, :]$ denotes the representation of node $i$, and $\sigma(\cdot)$ is the sigmoid function that maps the mask values $\mathbf{M}_i^x$ and $\mathbf{M}_{ij}^a$ to $[0, 1]$.

**Adversarial Causal Augmentation.** To estimate $\widetilde{g}^*$ in Equation 8, we define the adversarial learning objective as:

$$\max_{\theta_1} \{ \mathcal{L}_{\mathrm{adv}} = \mathbb{E}_{P_{\mathrm{tr}}} [\ell(f(T_{\theta_1}(g)), y) - \gamma c(T_{\theta_1}(g), g)] \}. \tag{10}$$

Then we can augment the graph by $T_{\theta_1}(g) = (\mathbf{A} \odot \mathbf{M}_{\mathrm{adv}}^a, \mathbf{X} \odot \mathbf{M}_{\mathrm{adv}}^x)$, where $\odot$ is the broadcasted element-wise product. Although adversarially augmented graphs guarantee environmental diversity,

it inevitably destroys the causal parts. Therefore, we utilize the causal generator $T_{\theta_2}(\cdot)$ to capture causal features and combine them with diverse environmental features. Following the sufficiency and independence conditions in Assumption 1, we define the causal learning objective as:

$$\min_{\theta, \theta_2} \{ \mathcal{L}_{\text{cau}} = \mathbb{E}_{P_{\text{tr}}}[\ell(f(T_{\theta_2}(g)), y) + \ell(f(\widetilde{g}), y)] \}, \tag{11}$$

where $\widetilde{g} = (\mathbf{A} \odot \widetilde{\mathbf{M}}^a, \mathbf{X} \odot \widetilde{\mathbf{M}}^x)$ is the augmented graph. It adopts the mask combination strategy: $\widetilde{\mathbf{M}}^a = (\mathbf{1}^a - \mathbf{M}_{\text{cau}}^a) \odot \mathbf{M}_{\text{adv}}^a + \mathbf{M}_{\text{cau}}^a$ and $\widetilde{\mathbf{M}}^x = (\mathbf{1}^x - \mathbf{M}_{\text{cau}}^x) \odot \mathbf{M}_{\text{adv}}^x + \mathbf{M}_{\text{cau}}^x$, where $\mathbf{M}_{\text{cau}}^a$ and $\mathbf{M}_{\text{cau}}^x$ are generated by $T_{\theta_2}(\cdot)$, $\mathbf{1}^a$ and $\mathbf{1}^x$ are all-one matrices, and if there is no edge between node $i$ and node $j$, then we set $\mathbf{1}_{ij}^a$ to 0.

Now we explain this combination strategy. Taking $\widetilde{\mathbf{M}}^x$ as an example, since $\mathbf{M}_{\text{cau}}^x$ denotes the captured causal regions via $T_{\theta_2}(\cdot)$, $\mathbf{1}^x - \mathbf{M}_{\text{cau}}^x$ represents the complementary parts, which are environmental regions. $\mathbf{M}_{\text{adv}}^x$ represents the adversarial perturbation, so $(\mathbf{1}^x - \mathbf{M}_{\text{cau}}^x) \odot \mathbf{M}_{\text{adv}}^x$ is equivalent to applying the adversarial perturbation on environmental features, meanwhile, sheltering the causal features. Finally, $+\mathbf{M}_{\text{cau}}$ indicates that the augmented data should retain the original causal features. Hence, this combination strategy achieves both environmental diversity and causal invariance. Inspecting Equation 11, the first term indicates that causal features are enough for predictions, thus satisfying the sufficiency condition. While the second term encourages causal features to make right predictions after perturbing the environments, thereby satisfying the independence condition.

**Regularization.** For Equation 10, the adversarial optimization tends to remove more nodes and edges, so we should also constrain the perturbations. Although Equation 11 satisfies the sufficiency and independence conditions, it is necessary to impose constraints on the ratio of the causal features to prevent trivial solutions. Hence, we first define the regularization function $r(\mathbf{M}, k, \lambda) = (\sum_{ij} \mathbf{M}_{ij}/k - \lambda) + (\sum_{ij} \mathbb{I}[\mathbf{M}_{ij} > 0]/k - \lambda)$, where $k$ is the total number of elements to be constrained, $\mathbb{I} \in \{0, 1\}$ is an indicator function. The first term penalizes the average ratio close to $\lambda$, while the second term encourages an uneven distribution. Given a graph with $n$ nodes and $m$ edges, we define the regularization term for adversarial augmentation and causal learning as:

$$\mathcal{L}_{\text{reg}_1} = \mathbb{E}_{P_{\text{tr}}}[r(\mathbf{M}_{\text{adv}}^x, n, \lambda_a) + r(\mathbf{M}_{\text{adv}}^a, m, \lambda_a)], \tag{12}$$

$$\mathcal{L}_{\text{reg}_2} = \mathbb{E}_{P_{\text{tr}}}[r(\mathbf{M}_{\text{cau}}^x, n, \lambda_c) + r(\mathbf{M}_{\text{cau}}^a, m, \lambda_c)], \tag{13}$$

where $\lambda_c \in (0, 1)$ is the ratio of causal features, we set $\lambda_a = 1$ for adversarial learning, which can alleviate excessive perturbations. The detailed algorithm of AdvCA is provided in Appendix A.1.

## 4 EXPERIMENTS

In this section, we conduct extensive experiments to answer the following **R**esearch **Q**uestions:

- **RQ1:** Compared to existing efforts, how does AdvCA perform under covariate shift?
- **RQ2:** Can the proposed AdvCA achieve the principles of environmental diversity and causal invariance, thereby effectively alleviating the covariate shift?
- **RQ3:** How do the different components of AdvCA affect performance?

### 4.1 EXPERIMENTAL SETTINGS

**Datasets.** We use graph OOD datasets (Gui et al., 2022) and OGB datasets (Hu et al., 2020), which include Motif, CMNIST, Molbbbp and Molhiv. Following Gui et al. (2022), we adopt the base, color, size and scaffold data splitting to create various covariate shifts. The details of the datasets, metrics and implementation details of AdvCA are provided in Appendix A.2 and A.3.

**Baselines.** We adopt 14 baselines, which can be divided into the following three specific categories:

- **Generalization Algorithms:** Empirical Risk Minimization (ERM), IRM (Arjovsky et al., 2019), GroupDRO (Sagawa et al., 2020), VREx (Krueger et al., 2021).
- **Graph Invariant Learning and Generalization:** DIR (Wu et al., 2022b), CAL (Sui et al., 2022), GSAT (Miao et al., 2022), OOD-GNN (Li et al., 2022a), StableGNN (Fan et al., 2021).
- **Graph Data Augmentation:** DropEdge (Rong et al., 2020), GREA (Liu et al., 2022), FLAG (Kong et al., 2022), M-Mixup (Wang et al., 2021), $\mathcal{G}$-Mixup (Han et al., 2022).

Table 1: Performance on synthetic and real-world datasets. Numbers in **bold** indicate the best performance, while the underlined numbers indicate the second best performance.

| Method | Motif | | CMNIST | Molbbbp | | Molhiv | |
|---|---|---|---|---|---|---|---|
| | base | size | color | scaffold | size | scaffold | size |
| ERM | $68.66_{\pm4.25}$ | $51.74_{\pm2.88}$ | $28.60_{\pm1.87}$ | $68.10_{\pm1.68}$ | $78.29_{\pm3.76}$ | $69.58_{\pm2.51}$ | $59.94_{\pm2.37}$ |
| IRM | $70.65_{\pm4.17}$ | $51.41_{\pm3.78}$ | $27.83_{\pm2.13}$ | $67.22_{\pm1.15}$ | $77.56_{\pm2.48}$ | $67.97_{\pm1.84}$ | $59.00_{\pm2.92}$ |
| GroupDRO | $68.24_{\pm8.92}$ | $51.95_{\pm5.86}$ | $29.07_{\pm3.14}$ | $66.47_{\pm2.39}$ | $79.27_{\pm2.43}$ | $70.64_{\pm2.57}$ | $58.98_{\pm2.16}$ |
| VREx | $\underline{71.47}_{\pm6.69}$ | $52.67_{\pm5.54}$ | $28.48_{\pm2.87}$ | $68.74_{\pm1.03}$ | $78.76_{\pm2.37}$ | $70.77_{\pm2.84}$ | $58.53_{\pm2.88}$ |
| DIR | $62.07_{\pm8.75}$ | $52.27_{\pm4.56}$ | $\underline{33.20}_{\pm6.17}$ | $66.86_{\pm2.25}$ | $76.40_{\pm4.43}$ | $68.07_{\pm2.29}$ | $58.08_{\pm2.31}$ |
| CAL | $65.63_{\pm4.29}$ | $51.18_{\pm5.60}$ | $27.99_{\pm3.24}$ | $68.06_{\pm2.60}$ | $\underline{79.50}_{\pm4.81}$ | $67.37_{\pm3.61}$ | $57.95_{\pm2.24}$ |
| GSAT | $62.80_{\pm11.41}$ | $53.20_{\pm8.35}$ | $28.17_{\pm1.26}$ | $66.78_{\pm1.45}$ | $75.63_{\pm3.83}$ | $68.66_{\pm1.35}$ | $58.06_{\pm1.98}$ |
| OOD-GNN | $61.10_{\pm7.87}$ | $52.61_{\pm4.67}$ | $26.49_{\pm2.94}$ | $66.72_{\pm1.23}$ | $79.48_{\pm4.19}$ | $70.46_{\pm1.97}$ | $60.60_{\pm3.77}$ |
| StableGNN | $57.07_{\pm14.10}$ | $46.93_{\pm8.85}$ | $28.38_{\pm3.49}$ | $66.74_{\pm1.30}$ | $77.47_{\pm4.69}$ | $68.44_{\pm1.33}$ | $56.71_{\pm2.79}$ |
| DropEdge | $45.08_{\pm4.46}$ | $45.63_{\pm4.61}$ | $22.65_{\pm2.90}$ | $66.49_{\pm1.55}$ | $78.32_{\pm3.44}$ | $\underline{70.78}_{\pm1.38}$ | $58.53_{\pm1.26}$ |
| GREA | $56.74_{\pm9.23}$ | $\underline{54.13}_{\pm10.02}$ | $29.02_{\pm3.26}$ | $\underline{69.72}_{\pm1.66}$ | $77.34_{\pm3.52}$ | $67.79_{\pm2.56}$ | $\underline{60.71}_{\pm2.20}$ |
| FLAG | $61.12_{\pm5.39}$ | $51.66_{\pm4.14}$ | $32.30_{\pm2.69}$ | $67.69_{\pm2.36}$ | $79.26_{\pm2.26}$ | $68.45_{\pm2.30}$ | $60.59_{\pm2.95}$ |
| M-Mixup | $70.08_{\pm3.82}$ | $51.48_{\pm4.91}$ | $26.47_{\pm3.45}$ | $68.75_{\pm0.34}$ | $78.92_{\pm2.43}$ | $68.88_{\pm2.63}$ | $59.03_{\pm3.11}$ |
| $\mathcal{G}$-Mixup | $59.66_{\pm7.03}$ | $52.81_{\pm6.73}$ | $31.85_{\pm5.82}$ | $67.44_{\pm1.62}$ | $78.55_{\pm4.16}$ | $70.01_{\pm2.52}$ | $59.34_{\pm2.43}$ |
| AdvCA (ours) | $\mathbf{73.64}_{\pm5.15}$ | $\mathbf{55.85}_{\pm7.98}$ | $\mathbf{36.37}_{\pm4.44}$ | $\mathbf{70.79}_{\pm1.53}$ | $\mathbf{81.03}_{\pm5.15}$ | $\mathbf{71.15}_{\pm1.81}$ | $\mathbf{61.64}_{\pm3.37}$ |
| Improvement | ↑ 2.17% | ↑ 1.72% | ↑ 3.17% | ↑ 1.07% | ↑ 1.53% | ↑ 0.37% | ↑ 0.93% |

## 4.2 MAIN RESULTS (RQ1)

To demonstrate the superiority of AdvCA, we first make comprehensive comparisons with baseline methods. The implementation settings and details of baselines are provided in Appendix A.4. All experimental results are summarized in Table 1. We have the following **Obs**ervations.

**Obs1: Most generalization and augmentation methods fail under covariate shift.** Generalization and data augmentation algorithms perform well on certain datasets or shifts. VREx achieves a 2.81% improvement on Motif (base). For two shifts of Molhiv, data augmentation methods GREA and DropEdge obtain 1.20% and 0.77% improvements. The invariant learning methods DIR and CAL also obtain 4.60% and 1.53% improvements on CMNIST and Molbbbp (size). Unfortunately, none of the methods consistently outperform ERM. For example, GREA and DropEdge perform poorly on Motif (base), ↓11.92% and ↓23.58%. DIR and CAL also fail on Molhiv. These show that both invariant learning and data augmentation methods have their own weaknesses, which lead to unstable performance when facing complex and diverse covariate shifts from different datasets.

**Obs2: AdvCA consistently outperforms all baseline methods.** Compared with ERM, AdvCA can obtain significant improvements. For two types of covariate shifts on Motif, AdvCA surpasses ERM by 4.98% and 4.11%, respectively. In contrast to the large performance variances on different datasets achieved by baselines, AdaCA consistently obtains the leading performance across the board. For CMNIST, AdvCA achieves a performance improvement of 3.17% compared to the best baseline DIR. For Motif, the performance is improved by 2.17% and 1.72% compared to VREx and GREA. These results illustrate that AdvCA can overcome the shortcomings of invariant learning and data augmentation. Armed with the principles of environmental diversity and causal invariance, AdvCA achieves stable and consistent improvements on different datasets with various covariate shifts. In addition, although we focus on covariate shift in this work, we also carefully check the performance of AdvCA under correlation shift, and the results are presented in Appendix D.1.

## 4.3 COVARIATE SHIFT AND VISUALIZATIONS (RQ2)

In this section, we conduct quantitative experiments to demonstrate that AdvCA can shorten the distribution gap, as shown in Figure 1. Specifically, we utilize $\mathrm{GCS}(\cdot,\cdot)$ as the measurement to quantify the degree of covariate shift. The detailed estimation procedure is provided in Appendix B. To make comprehensive comparisons, we select four different types of covariate shift: base, size, color and scaffold, to conduct experiments. We choose three data augmentation baselines, DropEdge, FLAG and $\mathcal{G}$-Mixup, which augment graphs from different views. The experimental results are shown in Table 2. We calculated covariate shifts between the augmentation distribution $P_{\mathrm{aug}}$ with the training $P_{\mathrm{tr}}$ or test distribution $P_{\mathrm{te}}$. "Aug-Train" and "Aug-Test" represent $\mathrm{GCS}(P_{\mathrm{aug}}, P_{\mathrm{tr}})$ and $\mathrm{GCS}(P_{\mathrm{aug}}, P_{\mathrm{te}})$, respectively. From the results in Table 2, we have the following observations.

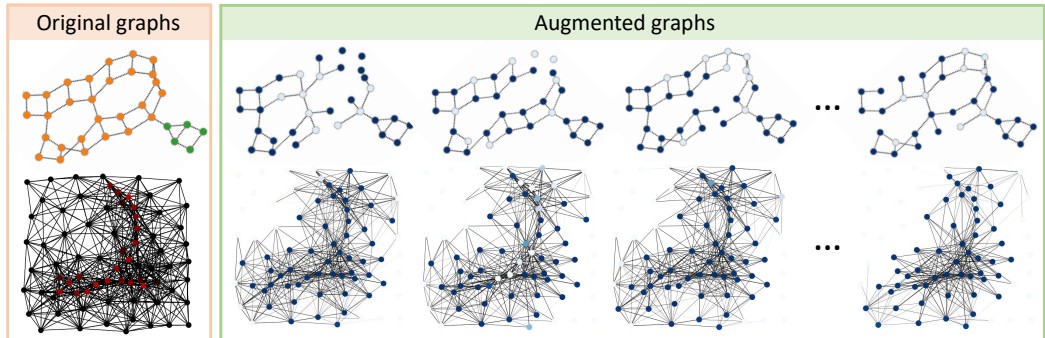

Figure 3: Visualizations of the augmented graphs via AdvCA.

Table 2: Covariate shift comparisons with different augmentation strategies.

| Method | Motif (base) | | Motif (size) | | CMNIST (color) | | Molbbbp (scaffold) | |
|---|---|---|---|---|---|---|---|---|
| | Aug-Train | Aug-Test | Aug-Train | Aug-Test | Aug-Train | Aug-Test | Aug-Train | Aug-Test |
| Original | 0 | $0.557_{\pm0.141}$ | 0 | $0.522_{\pm0.421}$ | 0 | $0.490_{\pm0.226}$ | 0 | $0.419_{\pm0.079}$ |
| DropEdge | $0.772_{\pm0.213}$ | $0.515_{\pm0.033}$ | $0.851_{\pm0.138}$ | $0.161_{\pm0.271}$ | $0.627_{\pm0.186}$ | $0.539_{\pm0.260}$ | $0.758_{\pm0.192}$ | $0.737_{\pm0.211}$ |
| FLAG | $0.001_{\pm0.001}$ | $0.533_{\pm0.016}$ | $0.002_{\pm0.018}$ | $0.507_{\pm0.121}$ | $0.003_{\pm0.002}$ | $0.442_{\pm0.062}$ | $0.001_{\pm0.001}$ | $0.413_{\pm0.088}$ |
| $\mathcal{G}$-Mixup | $0.690_{\pm0.186}$ | $0.472_{\pm0.043}$ | $0.816_{\pm0.154}$ | $0.299_{\pm0.343}$ | $0.408_{\pm0.228}$ | $0.351_{\pm0.318}$ | $0.551_{\pm0.258}$ | $0.545_{\pm0.231}$ |
| AdvCA | $0.369_{\pm0.169}$ | $\mathbf{0.462_{\pm0.063}}$ | $0.649_{\pm0.143}$ | $\mathbf{0.098_{\pm0.070}}$ | $0.516_{\pm0.106}$ | $\mathbf{0.307_{\pm0.108}}$ | $0.422_{\pm0.049}$ | $\mathbf{0.393_{\pm0.028}}$ |

**Obs3: AdvCA effectively closes the distribution gap with test distribution.** "Original" represents the original training distribution without augmentation. We observe that there exist large covariate shifts between the training and test distributions, ranging from 0.419 to 0.557. DropEdge greatly enlarges *Aug-Train*, *i.e.,* 0.627~0.851. While it fails to reduce *Aug-Test*, *e.g.,* on CMNIST (0.490→0.539) and Molbbbp (0.419→0.737). FLAG only perturbs the node features, leading to small values in *Aug-Train* and an inability to reduce *Aug-Test*. $\mathcal{G}$-Mixup significantly increases *Aug-Train* by generating OOD samples, while it cannot guarantee a decrease in *Aug-Test*. Finally, our proposed AdvCA enlarges the gap with training distribution by augmenting the environmental features. Meanwhile, the invariance of causal features significantly reduces *Aug-Test*, *e.g.,* on Motif-base (0.557→0.462), Motif-size (0.522→0.098) and CMNIST (0.490→0.408).

To verify the environmental diversity and causal invariance of AdvCA, we plot the augmented graphs in Figure 3. These augmented graphs are randomly sampled during training. More visualizations are depicted in Appendix D.3. The Motif and CMNIST graphs are displayed in the first and second rows. Figure 3 (*Left*) shows the original graphs. For Motif, the green part represents the motif-graph, whose type determines the label. While the yellow part denotes the base-graph that contains environmental features. For CMNIST, the red subgraph contains causal features while the complementary parts contain environmental features. Figure 3 (*Right*) displays the augmented samples during training. Nodes with darker colors and edges with wider lines indicate higher soft-mask values. From these visualizations, we have the following observations.

**Obs4: AdvCA can achieve both environmental diversity and causal invariance.** We can observe that AdvCA only perturbs the environmental features while keeping the causal parts invariant. For Motif dataset, the base-graph is a *ladder* and the motif-graph is a *house*. After augmentation, the nodes and edges of the *ladder* graph are perturbed. In contrast, the *house* part remains invariant and stable during training. The CMNIST graph also exhibits the same phenomenon. The environmental features are frequently perturbed, while the causal subgraph that determines label "2" remains invariant and stable during training. These visualizations further demonstrate that AdvCA can simultaneously guarantee environmental diversity and causal invariance.

## 4.4 ABLATION STUDY (RQ3)

**Adversarial augmentation v.s. Causal learning.** They are two vital components that achieve environmental diversity and causal invariance. The results are depicted in Figure 4 (*Left*). "w/o Adv" and "w/o Cau" refer to AdvCA without adversarial augmentation and without causal learning, respectively. RDCA stands for a variant that replaces the adversarial augmentation in AdvCA with

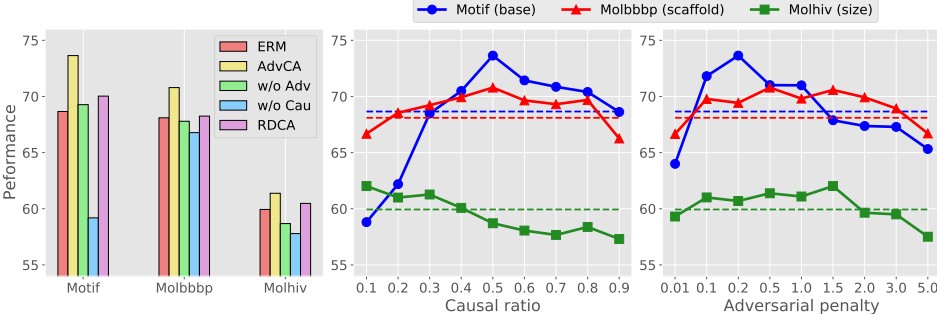

Figure 4: (*Left*): Performance comparisons of different components in AdvCA. (*Middle*): Performance over different causal ratios $\lambda_c$. (*Right*): Performance over different penalties $\gamma$.

random augmentation (*i.e.,* random masks). Compared to AdvCA, utilizing either causal learning or adversarial augmentation alone will degrade the performance. On the one hand, removing adversarial perturbations loses the invariance condition in causal learning, leading to suboptimal causal features. On the other hand, using adversarial augmentation alone will destroy the causal features, thereby impairing generalization. RDCA exceeds ERM, but is worse than AdvCA, suggesting that randomness will also encourage diversity, even if it is less effective than the adversarial strategy.

**Sensitivity Analysis.** The causal ratio $\lambda_c$ and penalty coefficient $\gamma$ determine the extent of causal features and the strength of adversarial augmentation, respectively. We also study their sensitivities. The experimental results are shown in Figure 4 (*Middle*) and (*Right*). Dashed lines denote the performance of ERM. $\lambda_c$ with 0.3~0.8 performs well on Motif and Molbbbp, while Molhiv is better in 0.1~0.3. It indicates that the causal ratio is a dataset-sensitive hyper-parameter that needs careful tuning. For the penalty coefficient, the appropriate values on the three datasets range from 0.1~1.5.

## 5 RELATED WORK

**Invariant Causal Learning** (Lu et al., 2021) exploits causal features for better generalization. IRM (Arjovsky et al., 2019) minimizes the empirical risks within different environments. Chang et al. (2020) minimize the performance gap between environment-aware and environment-agnostic predictors to discover rationales. Motivated by these efforts, DIR (Wu et al., 2022b) constructs multiple interventional environments for invariant learning. GREA (Liu et al., 2022) and CAL (Sui et al., 2022) learn causal features by challenging different environments. However, they only focus on correlation shift issues. The limited environments hinder their successes on covariate shift.

**Graph Data Augmentation** (Ding et al., 2022; Zhao et al., 2022; Yoo et al., 2022) enlarges the training distribution by perturbing features in graphs. Recent studies (Ding et al., 2021; Wiles et al., 2022) observe that it often outperforms other generalization efforts (Arjovsky et al., 2019; Sagawa et al., 2020). DropEdge (Rong et al., 2020) randomly removes edges, while FLAG (Kong et al., 2022) augments node features with an adversarial strategy. M-Mixup (Wang et al., 2021) interpolates graphs in semantic space. However, studies (Arjovsky et al., 2019; Lu et al., 2021) point out that causal features are the key to OOD generalization. These augmentation efforts are prone to perturb the causal features, which easily loses control of the perturbed distributions. Due to the space constraints, we put more discussions about OOD generalization in Appendix F.

## 6 CONCLUSION & LIMITATIONS

In this work, we focus on the graph generalization problem under covariate shift, which is of great need but largely unexplored. We propose a novel graph augmentation strategy, AdvCA, which is based on the principle of environmental diversity and causal invariance. Environmental diversity allows the model to explore more novel environments, thereby better generalizing to possible unseen test distributions. Causal invariance closes the distribution gap between the augmented and test data, resulting in better generalization. We make comprehensive comparisons with 14 baselines and conduct in-depth analyses and visualizations. The experimental results demonstrate that AdvCA can achieve excellent generalization ability under covariate shift. In addition, we also provide more discussions about the limitations of AdvCA and future work in Appendix G.

ETHICS STATEMENT

This paper does not involve any human subjects and does not raise any ethical concerns. We propose a graph data augmentation method to address the OOD issue of covariate shift. We conduct experiments on public datasets and validate the effectiveness on graph classification tasks. Our proposed method can be applied to practical applications, such as the prediction of molecular properties.

REPRODUCIBILITY STATEMENT

To help researchers reproduce our results, we provide detailed instructions here. For the implementation process of AdvCA, we provide the detailed algorithm in Appendix A.1. For datasets, we use publicly available datasets, and a detailed introduction is provided in Appendix A.2. For training details, we provide training settings of AdvCA and baseline settings in Appendix A.3 and A.4, respectively. Furthermore, we also provide an anonymous code link: https://anonymous.4open.science/r/AdvCA-68BF

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

# A  IMPLEMENTATION DETAILS

## A.1  ALGORITHM

We summarize the detailed implementations of AdvCA in Algorithm 1. Inspired by Suresh et al. (2021), we alternately optimize the adversarial augmenter and causal generator with the backbone model, in lines 13 and 14. We adopt the causal features for predictions in the inference stage.

---

**Algorithm 1:** Adversarial Causal Augmentation

---

**Require:** Training set $\mathcal{D}_{\text{tr}}$; Adversarial augmenter $T_{\theta_1}(\cdot)$; Causal generator $T_{\theta_2}(\cdot)$; GNN classifier $f(\cdot)$ with parameters $\theta$; Learning rates $\alpha, \beta$; Batch size $N$; Causal ratio $\lambda_c$; Penalty $\gamma$.

1: Randomly initilize $\theta, \theta_1, \theta_2$
2: **while** not converge **do**
3:     Sample a batch $\mathcal{B}_{\text{tr}} \leftarrow \{(g_i, y_i)\}_{i=1}^N \subset \mathcal{D}_{\text{tr}}$
4:     **for** each $(g_i, y_i) \in \mathcal{B}_{\text{tr}}$ **do**
5:         $\mathbf{M}_{\text{adv}}^a, \mathbf{M}_{\text{adv}}^x \leftarrow T_{\theta_1}(g_i)$ // adversarial perturbations
6:         $\mathbf{M}_{\text{cau}}^a, \mathbf{M}_{\text{cau}}^x \leftarrow T_{\theta_2}(g_i)$ // regions of causal features
7:         $\widetilde{\mathbf{M}}^a \leftarrow (\mathbf{1} - \mathbf{M}_{\text{cau}}^a) \odot \mathbf{M}_{\text{adv}}^a + \mathbf{M}_{\text{cau}}^a$ // augment edges
8:         $\widetilde{\mathbf{M}}^x \leftarrow (\mathbf{1} - \mathbf{M}_{\text{cau}}^x) \odot \mathbf{M}_{\text{adv}}^x + \mathbf{M}_{\text{cau}}^x$ // augment nodes
9:         $\widetilde{g}_i \leftarrow (\mathbf{A}_i \odot \widetilde{\mathbf{M}}^a, \mathbf{X}_i \odot \widetilde{\mathbf{M}}^x)$ // augmented graph
10:    **end for**
11:    Compute $\mathcal{L}_{\text{adv}} - \mathcal{L}_{\text{reg}_1}$ via Equation 10 and Equation 12
12:    Compute $\mathcal{L}_{\text{cau}} + \mathcal{L}_{\text{reg}_2}$ via Equation 11 and Equation 13
13:    Update parameters of adversarial augmenter via gradient ascent:
        $\theta_1 \leftarrow \theta_1 + \alpha \nabla_{\theta_1}(\mathcal{L}_{\text{adv}} - \mathcal{L}_{\text{reg}_1})$
14:    Update parameters of GNN and causal generator via gradient descent:
        $\theta \leftarrow \theta - \beta \nabla_\theta(\mathcal{L}_{\text{cau}} + \mathcal{L}_{\text{reg}_2}); \theta_2 \leftarrow \theta_2 - \beta \nabla_{\theta_2}(\mathcal{L}_{\text{cau}} + \mathcal{L}_{\text{reg}_2})$
15: **end while**

---

## A.2  DATASETS AND METRICS

**Datasets.** In this paper, we conduct experiments on graph OOD datasets (Gui et al., 2022) and OGB datasets (Hu et al., 2020), which include Motif, CMNIST, Molbbbp and Molhiv. We follow Gui et al. (2022) to create various covariate shifts, according to base, color, size and scaffold splitting. Base, color, size and scaffold are features of the graph data and do not determine the labels of the data, so they can be regarded as environmental features. The statistics of the datasets are summarized in Table 3. Below we give a brief introduction to each dataset.

- **Motif:** It is a synthetic dataset from Spurious-Motif (Wu et al., 2022b; Sui et al., 2022). As shown in original graphs in Figure 5, each graph is composed of a base-graph (*wheel, tree, ladder, star, path*) and a motif (*house, cycle, crane*). The label is only determined by the type of motif. We create covariate shift according to the base-graph type and the graph size (*i.e.,* node number). For base covariate shift, we adopt graphs with *wheel, tree, ladder* base-graphs for training, *star* for validation and *path* for testing. For size covariate shift, we use small-size of graphs for training, while the validation and the test sets include the middle- and the large-size graphs, respectively.

- **CMNIST:** Color MNIST dataset contains graphs transformed from MNIST via superpixel techniques (Monti et al., 2017). We define color as the environmental features to create the covariate shift. Specifically, we color digits with 7 different colors, where five of them are adopted for training while the remaining two are used for validation and testing.

- **Molbbbp & Molhiv:** These are molecular datasets collected from MoleculeNet (Wu et al., 2018). We define the scaffold and graph size (*i.e.,* node number) as the environmental features to create two types of covariate shifts. For scaffold shift, we follow (Gui et al., 2022) and use scaffold split to create training, validation and test sets. For size shift, we adopt the large-size of graphs for training and the smaller ones for validation and testing.

**Metrics.** We adopt classification accuracy as the metric for Motif and CMNIST. As suggested by Hu et al. (2020), we use ROC-AUC for Molhiv and Molbbbp datasets. In addition, we use $\text{GCS}(P, Q)$

Table 3: Statistics of graph classification datasets.

| Dataset | | Motif | | CMNIST | Molbbbp | | Molhiv | |
|---|---|---|---|---|---|---|---|---|
| Covariate shift | | base | size | color | scaffold | size | scaffold | size |
| Train | Graph# | 18000 | 18000 | 42000 | 1631 | 1633 | 24682 | 26169 |
| | Avg. node# | 17.07 | 16.93 | 75.00 | 22.49 | 27.02 | 26.25 | 27.87 |
| | Avg. edge# | 48.89 | 43.57 | 1392.76 | 48.43 | 58.71 | 56.68 | 60.20 |
| Val | Graph# | 3000 | 3000 | 7000 | 204 | 203 | 4113 | 2773 |
| | Avg. node# | 15.82 | 39.22 | 75.00 | 33.20 | 12.06 | 24.95 | 15.55 |
| | Avg. edge# | 33.00 | 107.03 | 1393.73 | 71.84 | 24.27 | 54.53 | 32.77 |
| Test | Graph# | 3000 | 3000 | 7000 | 204 | 203 | 4108 | 3961 |
| | Avg. node# | 14.97 | 87.18 | 75.00 | 27.51 | 12.26 | 19.76 | 12.09 |
| | Avg. edge# | 31.54 | 239.65 | 1393.60 | 59.75 | 24.87 | 40.58 | 24.87 |
| Class# | | 3 | 3 | 10 | 2 | 2 | 2 | 2 |

Table 4: Hyper-parameter details of AdvCA.

| Dataset | Motif | | CMNIST | Molbbbp | | Molhiv | |
|---|---|---|---|---|---|---|---|
| Covariate shift | base | size | color | scaffold | size | scaffold | size |
| Backbone (layer-hidden) | 4-300 | 4-300 | 4-300 | 4-64 | 4-32 | 4-128 | 4-128 |
| Augmenter (layer-hidden) | 2-300 | 2-300 | 2-300 | 2-64 | 2-32 | 2-128 | 2-128 |
| Generator (layer-hidden) | 2-300 | 2-300 | 2-300 | 2-64 | 2-32 | 2-128 | 2-128 |
| Epoch | 100 | 100 | 100 | 100 | 100 | 100 | 100 |
| Optimizer | Adam | Adam | Adam | Adam | Adam | Adam | Adam |
| Batch size | 512 | 512 | 512 | 32 | 128 | 32 | 512 |
| Learning rate $\alpha$ | 1e-3 | 1e-3 | 1e-3 | 1e-3 | 5e-3 | 1e-3 | 1e-2 |
| Learning rate $\beta$ | 5e-3 | 5e-3 | 5e-3 | 1e-3 | 5e-3 | 1e-2 | 1e-2 |
| Causal ratio $\lambda_2$ | 0.5 | 0.5 | 0.5 | 0.5 | 0.5 | 0.1 | 0.1 |
| Adversarial penalty $\gamma$ | 0.2 | 0.2 | 0.2 | 0.5 | 0.5 | 0.5 | 0.5 |

to measure the covariate shift between distributions $P$ and $Q$. For all experimental results, we perform 10 random runs and report the mean and standard derivations.

## A.3 TRAINING SETTINGS

We use the NVIDIA GeForce RTX 3090 (24GB GPU) to conduct all our experiments. To make a fair comparison, we adopt GIN (Xu et al., 2019) as the default architectures to conduct all experiments. We tune the hyper-parameters in the following ranges: $\alpha$ and $\beta \in \{0.01, 0.005, 0.001\}$; $\lambda_2 \in \{0.1, ..., 0.9\}$; $\gamma \in \{0.01, 0.1, 0.2, 0.5, 1.0, 1.5, 2.0, 3.0, 5.0\}$; batch size $\in \{32, 128, 256, 512\}$; hidden layers $\in \{32, 64, 128, 300\}$. The hyper-parameters of AdvCA are summarized in Table 4.

## A.4 BASELINE SETTINGS

For a more comprehensive comparison, we selected 14 baselines. In this section, we give a detailed introduction to the settings of these methods.

- For ERM, IRM (Arjovsky et al., 2019), GroupDRO (Sagawa et al., 2020), VREx (Krueger et al., 2021), and M-Mixup (Wang et al., 2021), we report the results from the study (Gui et al., 2022) by default and reproduce the missing results on Molbbbp.

- For DIR (Wu et al., 2022b), CAL (Sui et al., 2022), GSAT (Miao et al., 2022), DropEdge (Rong et al., 2020), GREA (Liu et al., 2022), FLAG (Kong et al., 2022) and $\mathcal{G}$-Mixup (Han et al., 2022), they provide source codes for the implementations. We adopt default settings from their source codes and detailed hyper-parameters from their original papers.

- For OOD-GNN (Li et al., 2022a) and StableGNN (Fan et al., 2021), their source codes are not publicly available. We reproduce them based on the codes of StableNet (Zhang et al., 2021).

- For RDCA in Section 4.4, it is a variant that replaces the adversarial augmentation in AdvCA with random augmentation. In our implementation, we use all-one matrices to create the initial node and edges masks. Then we randomly set 20% of nonzero elements to zero in these masks. Finally, we apply these masks to the graphs for random data augmentation. The process of causal learning is consistent with AdvCA.

## B    ESTIMATION OF GRAPH COVARIATE SHIFT

In this section, we elaborate on the implementation details of estimating the graph covariate shift. Without loss of generality, we start with the example of estimating the graph covariate shift between the training and test distributions. Given training set and test set $\mathcal{D}_{\mathrm{tr}}$ and $\mathcal{D}_{\mathrm{te}}$, they follow probability distribution functions $P_{\mathrm{tr}}$ and $P_{\mathrm{te}}$. The process of estimating $\mathrm{GCS}(P_{\mathrm{tr}}, P_{\mathrm{te}})$ is summarized in the following two steps:

- Firstly, it is intractable to directly estimate the distribution in graph space $\mathbb{G}$. Inspired by Ye et al. (2022), we can obtain the graph features and estimate the distribution in feature space $\mathbb{F}$. Specifically, given a sample, we train a binary GNN classifier $f$ to distinguish which distribution it comes from, where $f(\cdot) = \Phi \circ h$, $h(\cdot) : \mathbb{G} \to \mathbb{F}$ is a graph encoder, and $\Phi(\cdot) : \mathbb{F} \to \{0, 1\}$ is a binary classifier. Then we can adopt the pre-trained GNN encoder $h$ to extract graph features.

- Secondly, we prepare the features and estimate the distribution of the data via Kernel Density Estimation (KDE) (Parzen, 1962). Finally, we adopt the Monte Carlo Integration under importance sampling (Binder et al., 1993) to approximate the integrals in Definition 1.

We summarize these implementations in Algorithm 2. In lines 4 and 5, to avoid the label shift (Ye et al., 2022), we adopt sample reweighting to ensure the balance of each class.

---

**Algorithm 2:** Estimation of Graph Covariate shift

**Require:** Training dataset $\mathcal{D}_{\mathrm{tr}}$ and test dataset $\mathcal{D}_{\mathrm{te}}$; Batch size $N$; Loss function $\ell$; GNN
    $f = \Phi \circ h$; Importance sampling size $M$; Threshold $\epsilon$.
**Ensure:** Estimated covariate shift $\mathrm{GCS}(P_{\mathrm{tr}}, P_{\mathrm{te}})$.

1: Initialize parameters of $f$
2:  # Train a graph classifier
3: **while** not converge **do**
4:     Sample a batch $\mathcal{B}_{\mathrm{tr}} \leftarrow \{(g_i, y_i)\}_{i=1}^{N} \subset \mathcal{D}_{\mathrm{tr}}$ and relabel all $y_i \leftarrow 0$
5:     Sample a batch $\mathcal{B}_{\mathrm{te}} \leftarrow \{(g_i, y_i)\}_{i=1}^{N} \subset \mathcal{D}_{\mathrm{te}}$ and relabel all $y_i \leftarrow 1$
6:     $\mathcal{B} \leftarrow \mathcal{B}_{\mathrm{tr}} \cup \mathcal{B}_{\mathrm{te}}$
7:     **for** each $(g_i, y_i) \in \mathcal{B}$ **do**
8:         Compute loss $\ell(f(g_i), y_i)$ and back-propagate gradients
9:     **end for**
10:    Update the parameters of $f$ via gradient descent and reset the gradients
11: **end while**
12: # Prepare the features for the estimation
13: Extract training and test feature sets $\mathcal{F}_{\mathrm{tr}}$ and $\mathcal{F}_{\mathrm{te}}$ via encoder $h$
14: $\mathcal{F} \leftarrow \mathcal{F}_{\mathrm{tr}} \cup \mathcal{F}_{\mathrm{te}}$
15: Scale $\mathcal{F}$ to zero mean and unit variance
16: $\hat{\omega} \leftarrow$ fit by KDE the distribution of $\mathcal{F}$
17: Split $\mathcal{F}$ to recover the original partition $\mathcal{F}'_{\mathrm{tr}}, \mathcal{F}'_{\mathrm{te}}$
18: $\hat{P}_{\mathrm{tr}}, \hat{P}_{\mathrm{te}} \leftarrow$ fit by KDE the distributions of $\mathcal{F}'_{\mathrm{tr}}, \mathcal{F}'_{\mathrm{te}}$
19: # Estimate the covariate shift
20: Initialize $\mathrm{GCS}(P_{\mathrm{tr}}, P_{\mathrm{te}}) \leftarrow 0$
21: **for** $t \leftarrow \{1, ..., M\}$ **do**
22:    $z \leftarrow$ sample from $\hat{\omega}$
23:    **if** $\hat{P}_{\mathrm{tr}}(z) < \epsilon$ or $\hat{P}_{\mathrm{te}}(z) < \epsilon$ **then**
24:       $\mathrm{GCS}(P_{\mathrm{tr}}, P_{\mathrm{te}}) \leftarrow \mathrm{GCS}(P_{\mathrm{tr}}, P_{\mathrm{te}}) + |\hat{P}_{\mathrm{tr}}(z) - \hat{P}_{\mathrm{te}}(z)| / \hat{\omega}(z)$
25:    **end if**
26: **end for**
27: $\mathrm{GCS}(P_{\mathrm{tr}}, P_{\mathrm{te}}) \leftarrow \mathrm{GCS}(P_{\mathrm{tr}}, P_{\mathrm{te}}) / 2M$

---

## C  CORRELATION SHIFT & COVARIATE SHIFT

From Assumption 1, we can observe that environmental features easily change outside the training distribution, owing to their noncausal nature. Hence, distribution shifts are only caused by the environmental features. Specifically, we define the joint distribution of training and test data as $P_{\text{tr}}(G, Y)$ and $P_{\text{te}}(G, Y)$, respectively. Since their joint distribution can be rewritten as $P_{\text{tr}}(G, Y) = P_{\text{tr}}(Y|G)P_{\text{tr}}(G)$ and $P_{\text{te}}(G, Y) = P_{\text{te}}(Y|G)P_{\text{te}}(G)$, we can find that there exist two main reasons that lead to distribution shift $P_{\text{tr}}(G, Y) \neq P_{\text{te}}(G, Y)$.

- **Correlation shift** $P_{\text{tr}}(Y|G) \neq P_{\text{te}}(Y|G)$**.** If the statistical correlation of environmental features and labels is inconsistent in training and test data, a well-fitted model in training data may fail in test data, which is also known as spurious correlation or correlation shift (Ye et al., 2022). Formally, correlation shift describes the conditional distribution $P_{\text{tr}}(Y|G) \neq P_{\text{te}}(Y|G)$.

- **Covariate shift** $P_{\text{tr}}(G) \neq P_{\text{te}}(G)$**.** If there exist environmental features in the test distribution that the model has not seen during training, it will also result in performance drop. This unseen distribution shift is well known as covariate shift (Gui et al., 2022). It means that the environmental features in test data are unseen in training data, which leads to $P_{\text{tr}}(G) \neq P_{\text{te}}(G)$. Hence, in Assumption 1, we quantitatively measure the covariate shift between $P_{\text{tr}}(G)$ and $P_{\text{te}}(G)$.

## D  MORE EXPERIMENTAL RESULTS

### D.1  RESULTS ON CORRELATION SHIFT

Although this work focuses on the OOD issue of covariate shift, for completeness, we also evaluate the performance of AdvCA under correlation shift. Following Gui et al. (2022), we choose three graph OOD datasets (*i.e.,* Motif, CMNIST, Molhiv) with three different graph features (*i.e.,* base, color, size) to create correlation shifts. For baselines, we choose three generalization algorithms (*i.e.,* ERM, IRM (Arjovsky et al., 2019), VREx (Krueger et al., 2021)), three graph generalization methods (*i.e.,* DIR (Wu et al., 2022b), CAL (Sui et al., 2022), OOD-GNN (Li et al., 2022a)) and three data augmentation methods (*i.e.,* DropEdge (Rong et al., 2020), FLAG

Table 5: Performance comparisons on synthetic and real-world datasets with correlation shift.

| Method | Motif | CMNIST | Molhiv |
|---|---|---|---|
| ERM | $81.44_{\pm 2.54}$ | $42.87_{\pm 1.37}$ | $63.26_{\pm 1.25}$ |
| IRM | $80.71_{\pm 2.81}$ | $42.80_{\pm 1.62}$ | $59.90_{\pm 1.17}$ |
| VREx | $81.56_{\pm 2.14}$ | $43.31_{\pm 1.03}$ | $60.23_{\pm 1.60}$ |
| DIR | $82.25_{\pm 2.15}$ | $44.87_{\pm 1.56}$ | $64.65_{\pm 1.34}$ |
| CAL | $81.94_{\pm 1.20}$ | $41.82_{\pm 0.85}$ | $62.36_{\pm 1.42}$ |
| OOD-GNN | $80.22_{\pm 2.28}$ | $39.03_{\pm 1.24}$ | $57.49_{\pm 1.08}$ |
| DropEdge | $78.97_{\pm 3.41}$ | $38.43_{\pm 1.94}$ | $54.92_{\pm 1.73}$ |
| FLAG | $80.91_{\pm 1.04}$ | $43.41_{\pm 1.38}$ | $66.44_{\pm 2.32}$ |
| M-Mixup | $77.63_{\pm 1.12}$ | $40.96_{\pm 1.21}$ | $64.87_{\pm 1.36}$ |
| AdvCA (ours) | $\mathbf{82.51_{\pm 2.81}}$ | $\mathbf{49.73_{\pm 1.70}}$ | $\mathbf{68.11_{\pm 1.82}}$ |

(Kong et al., 2022), M-Mixup (Wang et al., 2021)). The experimental results are shown in Table 5. We can observe that AdvCA can also effectively alleviate the correlation shift. These results demonstrate that AdvCA learns better causal features by encouraging environmental diversity, which can effectively break spurious correlations that are hidden in the training data.

### D.2  RESULTS ON COMMONLY USED DATASETS

To demonstrate the effectiveness of the proposed AdvCA, we also conduct experiments on commonly used TU datasets (Morris et al., 2020), which include MUTAG, NCI1, PROTEINS, COLLAB, IMDB-B, IMDB-M. These are real world datasets and have negligible distribution shift. For training settings, we follow CAL (Sui et al., 2022) and adopt GIN (Xu et al., 2019) as our backbone model. The experimental results are shown in Table 6. For the results, we can observe that our method can achieve the best performance over different datasets.

Table 6: Performance comparisons on TU datasets.

| Method | MUTAG | NCI1 | PROTEINS | COLLAB | IMDB-B | IMDB-M |
|---|---|---|---|---|---|---|
| ERM | $89.42_{\pm 7.40}$ | $82.71_{\pm 1.52}$ | $76.21_{\pm 3.83}$ | $82.08_{\pm 1.51}$ | $73.40_{\pm 3.78}$ | $51.53_{\pm 2.97}$ |
| CAL | $89.91_{\pm 8.34}$ | $83.89_{\pm 1.93}$ | $76.92_{\pm 3.31}$ | $82.68_{\pm 1.25}$ | $74.13_{\pm 5.21}$ | $52.60_{\pm 2.36}$ |
| DropEdge | $86.11_{\pm 9.41}$ | $82.35_{\pm 3.77}$ | $74.40_{\pm 3.10}$ | $80.59_{\pm 2.14}$ | $72.34_{\pm 5.83}$ | $51.06_{\pm 3.04}$ |
| FLAG | $89.45_{\pm 7.20}$ | $82.67_{\pm 2.12}$ | $76.89_{\pm 3.66}$ | $82.48_{\pm 1.79}$ | $73.37_{\pm 4.94}$ | $52.16_{\pm 2.70}$ |
| M-Mixup | $89.83_{\pm 7.67}$ | $83.89_{\pm 2.38}$ | $76.76_{\pm 3.40}$ | $82.90_{\pm 1.43}$ | $74.07_{\pm 4.76}$ | $52.89_{\pm 2.84}$ |
| AdvCA (ours) | $\mathbf{90.34}_{\pm 7.75}$ | $\mathbf{84.12}_{\pm 2.64}$ | $\mathbf{77.92}_{\pm 3.72}$ | $\mathbf{82.98}_{\pm 1.76}$ | $\mathbf{74.23}_{\pm 5.10}$ | $\mathbf{53.02}_{\pm 2.76}$ |

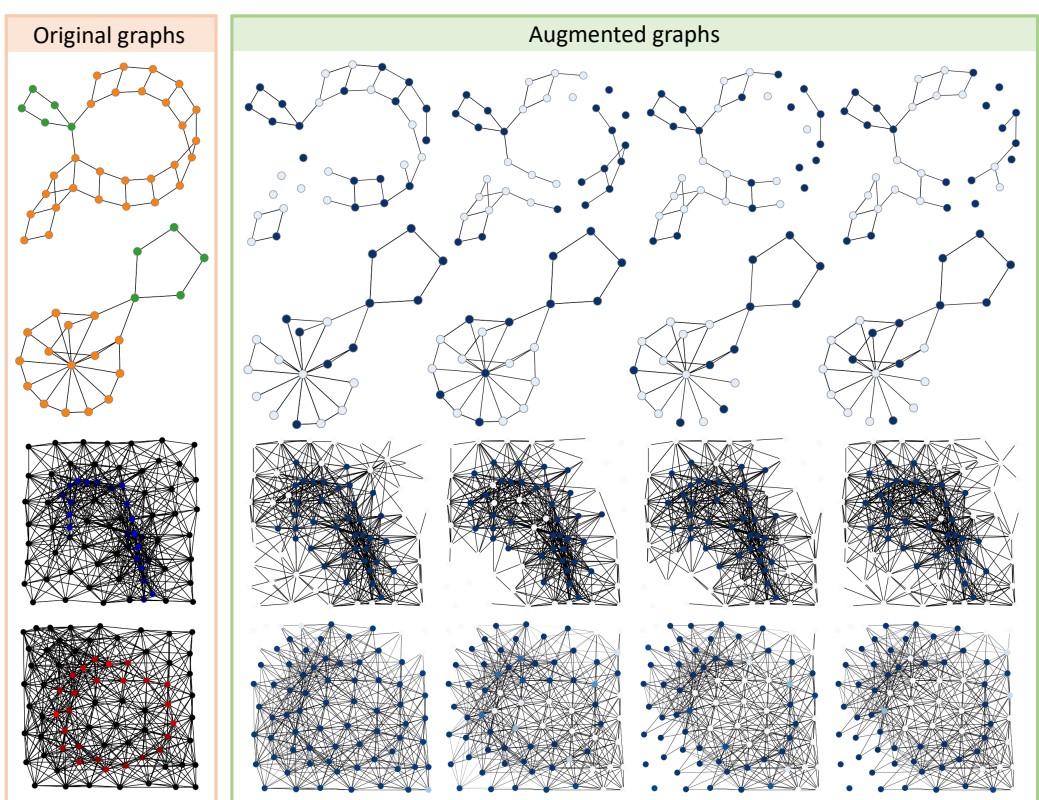

Figure 5: Visualizations of the augmented graphs via AdvCA.

### D.3 MORE VISUALIZATIONS

We display more visualizations of the augmented graph via AdvCA in Figure 5. To demonstrate the superiority our method, we also visualize the captured causal features by AdvCA and compare with other baselines. The results are displayed in Figure 6. From the results, we can easily observe that our method can find causal parts more accurately than other baseline methods.

## E COMPLEXITY ANALYSES

Firstly, we define the average numbers of nodes and edges per graph in the dataset to be $n$ and $m$, respectively. Let $N$ denote the batch size, $l$, $l_a$ and $l_c$ denote the numbers of layers in the GNN backbone, adversarial augmenter and causal generator, respectively. $d$, $d_a$ and $d_c$ are the dimensions of hidden layers in the GNN backbone, adversarial augmenter and causal generator, respectively.

**Time complexity.** The time complexity of the adversarial learning objective is $\mathcal{O}(N(l_a m d_a + 2lmd))$. For the causal learning objective, the time complexity is $\mathcal{O}(N(l_c m d_c + 2lmd))$. For the regularization terms, the time complexity is $\mathcal{O}(2N(n + m))$. For simplicity, we assume $l_a = l_c$ and $d_a = d_c$. Hence, the time complexity of a forward propagation is $\mathcal{O}(2N(l_a m d_a + 2lmd + n + m))$.

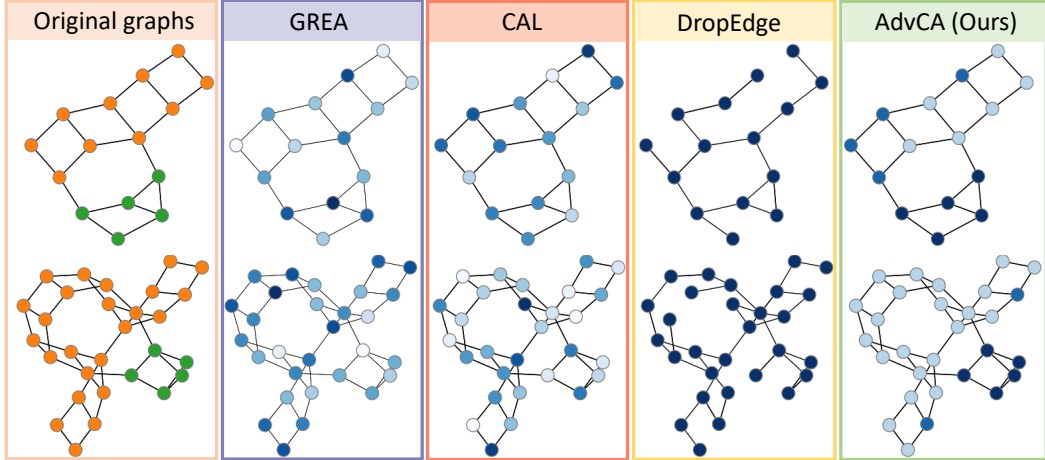

Figure 6: Visualization comparisons.

**Model size.** In addition to the GNN backbone model, we also introduce two small networks for adversarial augmentation and causal learning. In our implementations, the parameters of AdvCA are around twice as large as those of the original GNN model.

## F   MORE RELATED WORKS

**OOD Generalization** (Shen et al., 2021) has been widely explored. Recent studies (Ye et al., 2022; Gui et al., 2022; Wiles et al., 2022) point out that OOD falls into two specific categories: correlation shift and covariate shift. Correlation shift denotes that the environmental features and labels establish a statistical correlation that is inconsistent in training and test data. Thus, the models prefer to learn spurious correlations and rely on shortcut features (Geirhos et al., 2020) for predictions, resulting in a large performance drop. In contrast, covariate shift indicates that there exist unseen environmental features in test data. The limited training environment makes this issue intractable. In recent years, OOD generalization on graphs is drawing widespread attention (Li et al., 2022a;b; Fan et al., 2021; Wu et al., 2022a;b; Miao et al., 2022; Yu et al., 2022; Sui et al., 2022; Liu et al., 2022; Chen et al., 2022). However, these efforts mainly focus on correlation shift. While the issue of graph covariate shift is of great need but largely unexplored.

**Comprehensive Comparisons with EERM** (Wu et al., 2022a). Although EERM share similar goals with us, generating several environments through augmentation, there exist many technical and contribution differences. Firstly, EERM ignores the distinction between correlation shift and covariate shift problems, so it is not specifically designed for covariate shift. Different from them, we distinguish these two shifts in detail and design a novel framework specifically for covariate shift. Secondly, EERM does not model causal and environmental features, which results in the inability to explicitly distinguish them. In contrast, we explicitly model the environmental and causal features. Hence, we can effectively identify causal and environmental features and explicitly separate them from data. Thirdly, we also design a metric, $\mathrm{GCS}(\widetilde{P}, P)$, which can effectively measure the diversity of the environmental features for our augmented data. And we directly encourage the environmental diversity of the augmented samples by maximizing $\mathrm{GCS}(\widetilde{P}, P)$. However, EERM does not provide any evaluation metric for environmental diversity. To encourage the diversity, they "blindly" maximize the variance of the empirical risk in $K$ environments. Finally, for generalization scope, EERM is based on the IRM (Arjovsky et al., 2019) by minimizing the empirical risk in $K$ environments. In contrast, inspired by DRO (Sagawa et al., 2020), we can guarantee the generalization within the robust radius $\rho$. We summarize the above detailed discussions in Table 7.

## G   LIMITATION & FUTURE WORK

Although AdvCA outperforms numerous baselines and can achieve outstanding performance under various covariate shifts, we also prudently introspect the following limitations of our method. And we leave the improvements of these limitations as our future work.

Table 7: Comparisons with EERM.

| | | EERM | Our AdvCA |
|---|---|---|---|
| Scope | Is it specifically designed for covariate shift? | ✗ | ✓ |
| Separability | Can environmental/causal features be separated? | ✗ | ✓ |
| Environmental Diversity | Can environmental features be identified explicitly? | ✗ | ✓ |
| | How to model environmental features? | - | Mask model $T_{\theta_1}(\cdot)$ |
| | Metric for environmental diversity | - | $\text{GCS}(\widetilde{P}, P)$ |
| | Generation principle for environmental features | "Blindly" maximize $\mathbb{V}_e[R(e)]$ | Maximize $\text{GCS}(\widetilde{P}, P)$ |
| Causal Invariance | Can causal features be identified explicitly? | ✗ | ✓ |
| | How to model causal features? | - | Mask model $T_{\theta_2}(\cdot)$ |
| | Learning principles for causal features | $\min_\theta \mathbb{V}_e[R(e)]$ | Sufficiency/Independence |
| Generalization | Theoretical basis | IRM | DRO |
| | Generalization scope | $K$ environments | Robust radius $\rho$ $D(\widetilde{P}, P) \le \rho$ |

- AdvCA performs OOD exploration through an adversarial data augmentation strategy to achieve environmental diversity. However, it only perturbs the existing graph data in a given training set, such as perturbing original graph node features or graph structures. Hence, it is possible that there still exist some overlaps between the augmented distribution and training distribution, so Principle 1 cannot be thoroughly achieved. In future work, we will attempt to design more advanced data augmentation methods, such as graph generation-based strategies (Zhu et al., 2022), to generate more unseen and novel graph data, for pursuing Principle 1.

- For model training, we adopt adversarial training and causal learning to alternately optimize the adversarial augmenter, causal generator and backbone GNN. This training strategy may make the training process unstable, so the performance of AdvCA may experience a large variance. In addition, these two networks also involve additional parameters. Optimizing these parameters separately will also increase the time complexity, as shown in Appendix E. Hence, in future work, we will explore how to utilize more advanced optimization methods and lightweight models to achieve the principles of environmental diversity and causal invariance.

