# OpenReview forum: "Adversarial Causal Augmentation for Graph Covariate Shift"
_ICLR.cc/2023/Conference — Submitted to ICLR 2023_

### Official Review · Reviewer_rgVp · 2022-10-24

**Confidence:** 3
**Correctness:** 3
**Technical Novelty And Significance:** 3
**Empirical Novelty And Significance:** 3
**Recommendation:** 6

**Clarity, Quality, Novelty And Reproducibility:**

Clarity: Fair

Quality: Good

Novelty: Good

Reproducibility: Good

**Strength And Weaknesses:**

Strengths:
1. The paper focuses on the graph generalization problem under covariate shift, which is of great need but largely unexplored.
2. Comprehensive experiments with 14 baselines are conducted to demonstrate that AdvCA can achieve better generalization ability under covariate shift.

Weaknesses:
1. Experiments on more commonly used datasets like MUTAG and NCI1 are suggested to better demonstrate the effectiveness.
2. The example “ladder” and “tree” are absent in Figure 1, but are mentioned in paragraph 2, which make readers confused.
3. In Definition 1, since the prediction is independent of the environment features, you can somehow learn to distinguish the causal feature and environment features. Why don’t you only keep the causal feature to make a prediction?



**Summary Of The Paper:**

The paper focus on a specific type of OOD issue, covariate shift. The authors propose Adversarial Causal Augmentation (AdvCA) to alleviate the covariate shift. It adversarially augments the data to explore diverse distributions of the environments. Meanwhile, it keeps the causal features invariant across diverse environments. Extensive experimental results on synthetic and real-world datasets prove the effectiveness of AdvCA.

**Summary Of The Review:**

The paper focuses on the graph generalization problem under covariate shift, which is of great need but largely unexplored. However, there are some minor issues to be addressed.

---

> ### Author Response · Authors · 2022-11-12
> **Response to Reviewer rgVp**
>
> Thank you for your time and your valuable suggestions! We summarize your **3 concerns**, we provide the following responses. If you are satisfied, **we hope you can further improve our score**, many thanks!
>
> > **[Con 1: Experiments on commonly used datasets like MUTAG and NCI1]**
>
> Thanks, **common graph classification datasets, such as MUTAG and NCI1, are often considered as i.i.d datasets, thereby having negligible distribution shift.** Since our paper aims to address OOD issues, we adopt the OOD datasets.  But, to address your concerns, we also conducted experiments on commonly used datasets and compared to some baseline methods. The experimental results are shown in the following table. Experimental results show that our method can achieve the best performance over different datasets. We also provide these results in Appendix D.2 in our revised version, and please check them again.
>
> | Methods  |     MUTAG      |      NCI1      |    PROTEINS    |     COLLAB     |     IMDB-B     |     IMDB-M     |
> | :------- | :------------: | :------------: | :------------: | :------------: | :------------: | :------------: |
> | ERM      | 89.42$\pm$7.40 | 82.71$\pm$1.52 | 76.21$\pm$3.83 | 82.08$\pm$1.51 | 73.40$\pm$3.78 | 51.53$\pm$2.97 |
> | CAL      | 89.91$\pm$8.34 | 83.89$\pm$1.93 | 76.92$\pm$3.31 | 82.68$\pm$1.25 | 74.13$\pm$5.21 | 52.60$\pm$2.36 |
> | DropEdge | 86.11$\pm$9.41 | 82.35$\pm$3.77 | 74.40$\pm$3.10 | 80.59$\pm$2.14 | 72.34$\pm$5.83 | 51.06$\pm$3.04 |
> | FLAG     | 89.45$\pm$7.20 | 82.67$\pm$2.12 | 76.89$\pm$3.66 | 82.48$\pm$1.79 | 73.37$\pm$4.94 | 52.16$\pm$2.70 |
> | M-Mixup  | 89.83$\pm$7.67 | 83.89$\pm$2.38 | 76.76$\pm$3.40 | 82.90$\pm$1.43 | 74.07$\pm$4.76 | 52.89$\pm$2.84 |
> | Ours     | **90.34$\pm$7.75** | **84.12$\pm$2.64** | **77.92$\pm$3.72** | **82.98$\pm$1.76** | **74.23$\pm$5.10** | **53.02$\pm$2.76** |
>
> > **[Con 2: Confused about the “ladder” and “tree”]**
>
> Thank you for your suggestions. In our revised paper, we add detailed introductions about "ladder" and "tree" in Figure 1.
>
> > **[Con 3: Why don’t you only keep the causal feature to make a prediction?]**
>
> - **Inference stage.** Definition 1 (Assumption 1 in revised paper) is our assumption about the data. We define causal and environmental features following the assumption of existing studies on data generation process. **But we don't have ground-truths (i.e., location labels for real causal features) in training data, we need to let the model learn causal features by itself during model training.** Hence, we need to learn causal features via adversarial data augmentation. Yes, you are right. **Model only uses the learned causal features to make predictions in the inference stage. And we have stated this point in Appendix A.1**. Please check this in our original paper.  To facilitate your understanding, we also give detailed explanations of our training stage.
> - **Training stage.**  In model training state, we need to use both causal features and augmented environmental features as the model's input, and hope the model can make right prediction. This will encourage the model to learn how to distinguish causal features from data with more complex and variable environmental features. According to Definition 1, due to the invariance assumption of causal features, prediction based on causal features is the basis for the model to achieve OOD generalization.  Under covariate shift, there exist new environmental features in the test data that the model has not seen during training. This means that the model only learns in training data with limited environmental features, but need to generalize to test data with unseen environmental features. To solve this problem, **we use adversarial augmentation strategy to generate more data, which contains more diverse environmental features, to compensate for the scarcity of environmental features in the training data.** In this way, the model will see more diverse distributions of environmental features outside the training distribution. Hence, in the testing stage, the model can effectively and accurately identify causal features from data even with unseen environmental features, and make predictions only based on the causal features.
>
> Finally, thank you again for the valuable feedback! **If all your concerns have been resolved, we sincerely hope that you can further improve our score. Your support is very important to us, thank you very much!** Please let us know if you have any other comments/questions, we are more than happy to address them further!

---

> ### Author Response · Authors · 2022-12-05
> **Gentle reminder to the reviewer**
>
> Dear Reviewer rgVp,
>
> Thanks again for reviewing our paper. According to your suggestions, we have carefully addressed your concerns and provided additional experiments on commonly used datasets.
> As the discussion deadline is closing, we'd be grateful if you can confirm whether our responses have addressed your concerns. Please kindly let us know if there is anything else we can address to convince you for upgrading the scores. Thank you very much!
>
> Thanks.
>
> Authors.

---

### Official Review · Reviewer_5fRb · 2022-10-24

**Confidence:** 4
**Correctness:** 2
**Technical Novelty And Significance:** 2
**Empirical Novelty And Significance:** 3
**Recommendation:** 3

**Clarity, Quality, Novelty And Reproducibility:**

The reproducibility is good considering the codes are available, but I do have some concerns about the clarity, quality, and novelty as detailed above.

**Strength And Weaknesses:**

Pros:
1. The OOD generalization problem of graph machine learning is an important and trending research topic. I appreciate the authors' efforts in formalizing the problem and proposing necessary assumptions for graph covariate shifts.
2. The authors propose a causal framework to formulate the problem and guide the graph augmentation.
3.  Both synthetic and real datasets are used to show the empirical effectiveness of the proposed approach.

Cons:
However, I have some concerns:
1. The main concern is regarding technical novelty. I think the problem formulation is similar to previous works, e.g., EERM (Wu et al., ICLR 2022a), so the formulation should not be considered a particular contribution. The authors claim that the existing graph generalization efforts cannot guarantee both environmental diversity and causal invariance, but EERM has explored these key concepts and its formulation for ego-graphs can be straightforwardly extended to the graph-level tasks.
2. The proposed method is somehow incremental and thus the technical contribution of the paper is limited. In particular:
(1) The idea of adversarial data augmentation is similar to EERM. More comparisons and discussions regarding related works are expected to further clarify the novelty of the proposed method.
(2) The authors claim that the existing works cannot guarantee environmental diversity while the proposed AdvCA can. However, why AdvCA can increase diversity than augmentation-based baseline methods such as GREA is not clearly discussed.
3. The experiments are not convincing enough. The experimental results of baselines seem to be inconsistent with the results in the original papers. The details regarding the hyper-parameters of baselines are not clearly present, which raises potential concerns in fair comparisons. The visualizations should also consider some baselines to verify whether AdvCA finds causal parts more accurately than baselines.
4. I would also suggest the authors proofread Figure 2 since the plotted environmental features are identical and not diverse.



**Summary Of The Paper:**

This paper proposes a new method for out-of-distribution (OOD) generalization on graphs. The authors claim that the existing works mainly focus on the OOD issue of correlation shift, while another type, covariate shift, remains largely unexplored. Then, a graph augmentation strategy called AdvCA is proposed to handle the covariate shift problem with an adversarial augmenter and a causal generator. Experimental results show that the proposed AdvCA can outperform several baselines.

**Summary Of The Review:**

In summary, this paper focuses on an important problem and proposes an effective method to tackle this problem, but there are concerns regarding the technical novelty, clarity, and quality.

---

> ### Author Response · Authors · 2022-11-12
> **Response to Reviewer 5fRb (Part 3/3)**
>
>  $~~~~~~~~~~~~~~~~~~~~~~~~~~~~~~~~~~~~~~~~~~~~~~~~~~~~~~$ **Table S2: Graph classification results on EERM.**
>
> | Methods | Molbbbp (scaffold) | Molbbbp (size) |  Motif (base)  |  Motif (size)  |     CMNIST     | Molhiv (scaffold) | Molhiv (size)  |
> | :------ | :----------------: | :------------: | :------------: | :------------: | :------------: | :---------------: | -------------- |
> | ERM     |   68.10$\pm$1.68   | 78.29$\pm$3.76 | 68.66$\pm$4.25 | 51.74$\pm$2.88 | 28.60$\pm$1.87 |  69.58$\pm$2.51   | 59.94$\pm$2.37 |
> | EERM   |   66.32$\pm$4.87  | 73.48$\pm$7.85 |      OOM       |      OOM       |      OOM       |        OOM        | OOM            |
> | Ours    |   70.79$\pm$1.53   | 81.03$\pm$5.15 | 73.64$\pm$5.15 | 55.85$\pm$7.98 | 36.37$\pm$4.44 |  71.15$\pm$1.81   | 61.64$\pm$3.37 |
>
> > **[Con 4: Why AdvCA can increase diversity than GREA is not clearly discussed]**
>
> Thanks, we have repeatedly emphasized this point in our original paper (e.g. Introduction, Section 3.1 and Related work). GREA belongs to the paradigm of invariant learning, which implements data augmentation by combining causal features with different environmental features. However, **these environmental features are still derived from the given training set. In contrast, AdvCA can generate totally new environmental features outside the training distribution.** We have also demonstrated this experimentally.  Hence, our method can effectively increase the environmental diversity than GREA.
>
> > **[Con 5: Experiments are not convincing enough]**
>
> - **Inconsistent results and hyper-parameters of baselines.** The results are inconsistent with the original paper of the baselines, **because we use a completely different dataset GOOD** [5]. This dataset was just made public a few months ago and was just accepted by NeurIPS2022.  The reason we use this dataset is because we focus on the covariate shift, and there exist many covariate shift datasets in GOOD [5]. While most datasets used in baseline papers only contain correlation shift. For hyper-parameters of baselines, we have describe the all the settings in Appendix A.4. Specifically, for ERM, IRM, GroupDRO, VREx, Mixup and other methods, we directly report the results given in GOOD's original paper. For DIR, CAL, GSAT, DropEdge, GREA, FLAG, G-Mixup, we directly use original codes and default hyper-parameters provided by their original paper. For OOD-GNN and StableGNN, since they do not release code and share a similar idea with StableNet, we reproduce them based on StableNet. If the paper is accepted, we can promise to provide the complete code of all the baselines we reproduce in our paper.
> - **Visualizations comparisons with other baselines.** Thanks for your valuable suggestions, **we have added visualization results in the updated version.** We compare with GREA, CAL and DropEdge. From the results we can observe that our method can find causal parts more accurately than baselines. Due to the limited space, we put the results in Appendix D.3 (Figure 6).
>
> > **[Con 6: Proofread the Figure 2]**
>
> Thank you, maybe our drawing style cause a misunderstanding for you. The environmental features in the data are constantly changing during training. **We accept your suggestion and highlighted them in red color in our revised paper.**
>
> Finally, thank you again for the valuable feedback! **If all your concerns have been resolved, we sincerely hope that you can further improve our score. Your support is very important to us, thank you very much!** Please let us know if you have any other comments/questions, we are more than happy to address them further!
>
> [1] Invariant Risk Minimization
>
> [2] Invariance Principle Meets Information Bottleneck for Out-of-Distribution Generalization, NeurIPS 2021
>
> [3] Invariance Principle Meets Out-of-Distribution Generalization on Graphs
>
> [4] Out-Of-Distribution Generalization on Graphs: A Survey
>
> [5] GOOD: A Graph Out-of-Distribution Benchmark, NeurIPS 2022
>
> [6] OoD-Bench: Quantifying and Understanding Two Dimensions of Out-of-Distribution Generalization, CVPR 2022

---

> ### Author Response · Authors · 2022-11-12
> **Response to Reviewer 5fRb (Part 2/3)**
>
> >**[Con2: "EERM has explored these key concepts"; "Method is somehow incremental"; "Idea of adversarial data augmentation is similar to EERM"; "Technical contribution is limited".] (part 2)**
>
> In addition, **for implementations, we adopt totally different methods with EERM.** Specifically, **(1) Different augmentation methods.** EERM  introduces K additional data editors to generate K graphs based on the input graph.  It learns K graph editors by maximizing the variance of the risks, and encourages the model to minimize the variance.  In contrast, we adopt mask generation network to explore the data distribution as far as possible from the current training distribution, which makes efforts to jump out of the given distribution and generate more new training data with diverse environmental features. The environmental diversity encourages the model to learn how to distinguish causal features from data with more complex environmental features. **(2) Different optimization strategies.** EERM uses the REINFORCE algorithm to optimize the generator, which may cause a large variance in the gradient, while AdvCA uses the gradient ascent algorithm to optimize the mask generation network, so our optimization is more stable. In addition, we also list the following shortcomings in EERM:
>
> - **EERM does not show the environmental diversity of their augmented samples.** EERM does not show in any way the degree of environmental diversity of their generated data. In contrast, we fully demonstrate that the environmental features of data we generate is diverse from two perspectives. (1) From an intuitive and qualitative perspective, **we make visualization in Figure 3** to demonstrate that our method can indeed generate more complex and variable environmental features. (2) From a rigorous and quantitative perspective, **we quantitatively show that our method can indeed achieve more diverse environmental features in augmented data** by computing GCS in Table 2.
>
> - **EERM ignores the differences between correlation shift and covariate shift.** EERM does not specify exactly what type of distribution shift they focus on. In contrast, we capture the key difference between two distribution shifts, make comprehensive analyses for these shifts and design a specific solution for covariate shift.
> - **EERM does not provide a solution to measure the covariate shift.** In EERM, the distributional shift is only qualitatively defined by artificial transformation, time or region. **They don't quantitatively describe the distribution shift.** Therefore, it is unknown whether EERM can actually generate OOD data to reduce the gap between the test distribution. **In contrast, we propose an executable solution to quantitatively calculate the covariate shift**, which can effectively measure the gap between two graph data distributions. Hence, we also can proof that our method can effectively close the distribution gap with test distribution and alleviate the covariate shift issue.
>
> Finally, we humbly accept your advice and **add detailed comparisons and discussions with EERM in our revised version** (in Appendix F and Table 7).
>
> > **[Con3: EERM can be straightforwardly extended to the graph-level task.]**
>
> - **EERM entirely focuses on node classification task.** EERM does not give any implementation (e.g. algorithm or code) or experimental result for graph classification task. In contrast, a large body of recent efforts mainly focus on graph classification tasks, so we comprehensively summarize the common shortcomings of these methods to address the covariate shift.
> - **EERM has serious scalability issue and is difficult to transfer to graph-level tasks.** EERM separately designs K learnable matrices as generator for each graph data, so it requires O(K\*N\*N) memory consumption for each graph data, which has serious scalability issue. **This serious issue is also pointed by Reviewer xk81** in https://openreview.net/forum?id=FQOC5u-1egI. For a practical example, a recent NeurIPS2022 work GOOD [5] try to reproduce EERM on GOOD-Arxiv dataset, **but cause out-of-memory (OOM) issue**, and you can check this issue from their paper (in Table 2). This cumbersome design also makes EERM difficult to transfer to graph classification tasks. This is due to the large number of training data in graph classification tasks. For example, the training set of CMNIST has 42,000 graphs, and EERM needs to define K*42,000 learnable matrices in advance, which requires huge memory consumption. In contrast, our method just adopts lightweight mask generation network. Since all graph data share the same mask generation network, our method can effectively handle graph classification task. To address your concerns, we try to conduct experiments and the results are shown **in the following table (Table S2)**. We can find that, except for a small-scale dataset Molbbbp, all other datasets have (out-of-memory) OOM issues.

---

> ### Author Response · Authors · 2022-11-12
> **Response to Reviewer 5fRb (Part 1/3)**
>
> First of all, thanks for your carefully reading and valuable reviews!  We summarize your **6 concerns** and provide point-to-point responses. **We believe you will change your negative attitude and improve our score if you read them carefully!**
>
> > **[Con1: Similar problem formulation with EERM.]**
>
> Thanks. The problem formulation in our paper is only to clarify the problem to be solved and to facilitate the subsequent introduction of our method. We respectfully disagree that we are imitating EERM to define the problem. In addition, we can also find similar problem formulations in papers [1-4].
>
> > **[Con2: "EERM has explored these key concepts"; "Method is somehow incremental"; "Idea of adversarial data augmentation is similar to EERM"; "Technical contribution is limited".] (part 1)**
>
> Thanks for your valuable reviews! But we respectfully argue that our technical contribution is strong. And we argue that our AdvCA is significantly different from EERM w.r.t. the following aspects. The key differences are summarized **in the following table (Table S1)**.  In addition, **we also contacted the first author of EERM for a detailed discussion. The author agrees our contribution and the essential difference from EERM, and this table was also confirmed by the author of EERM.**
>
> ​		$~~~~~~~~~~~~~~~~~~~~~~~~~~~~~~~~~~~~~~~~~~~~~~~~~~~~~~~~~~~~~~~~~~~~~~~~~~~$				**Table S1: Comparisons with EERM.**
>
> |    $~~~~~~~~~~~~~$   Type  |        $~~~~~~~~~~~~~~~~~~~~~~~$     Comparisons  |         $~~~~~~~~~~~~~~~~~$     EERM    |      $~~~~~~~~~~~~~~$     Our AdvCA   |
> | :---------------------: | :--------------------------------------------------: | :-------------------------------------: | :----------------------------------------------------: |
> |          Scope          |   Is it specifically designed for covariate shift?   |   No     |                          Yes                           |
> |      Separability       |   Can environmental/causal features be separated?    |    No     |                          Yes                           |
> | Environmental Diversity | Can environmental features be identified explicitly? |                   No                    |                          Yes                           |
> | Environmental Diversity | How to model environmental features? |  -   |            Mask model $T_{\theta_1}(\cdot)$            |
> | Environmental Diversity | Metric for environmental diversity |  -  |             ${\rm GCS}(\widetilde{P}, P)$              |
> | Environmental Diversity | Generation principle for environmental features    | "Blindly" maximize $\mathbb{V}_e[R(e)]$ |         Maximize ${\rm GCS}(\widetilde{P}, P)$         |
> |    Causal Invariance    |    Can causal features be identified explicitly?     |      No    |      Yes   |
> |    Causal Invariance    |            How to model causal features?             |     -     |            Mask model $T_{\theta_2}(\cdot)$            |
> |    Causal Invariance    |       Learning principles for causal features        |  ${\rm min}_\theta\mathbb{V}_e[R(e)]$   |                Sufficiency/Independence                |
> |     Generalization      |   Theoretical basis                   |                   IRM                   |                          DRO                           |
> |     Generalization      |            Generalization scope            |            $K$ environments   | Robust radius $\rho$:  $D(\widetilde{P}, P) \leq \rho$ |
>
> - **Problem scope.** EERM ignores the distinction between correlation shift and covariate shift problems, so it is not specifically designed for covariate shift. Different from them, we distinguish these two shifts in detail and design a novel framework specifically for covariate shift.
>
> - **Separability of causal/environmental features.** EERM does not model causal and environmental features, which results in the inability to explicitly distinguish them. In contrast, we explicitly model the environmental and causal features.  Hence, we can effectively identify causal and environmental features and explicitly separate them from data.
>
> - **Environmental diversity.** We also design a metric, ${\rm GCS}(\widetilde{P}, P)$, which can effectively measure the diversity of the environmental features for our augmented data.  And we directly encourage the environmental diversity of the augmented samples by maximizing ${\rm GCS}(\widetilde{P}, P)$.  However, EERM does not provide any evaluation metric for environmental diversity. To encourage the diversity, they ”blindly“  maximize the variance of the empirical risk in $K$ environments.
>
> - **Generalization.** for generalization scope, EERM is based on the IRM by minimizing the empirical risk in $K$ environments. In contrast, inspired by DRO, we can guarantee the generalization within the robust radius $\rho$.

---

> ### Author Response · Authors · 2022-12-02
> **Sincerely expecting further discussions with you!**
>
> Dear Reviewer 5fRb:
>
> We really appreciated your time and your constructive reviews. We have made great efforts to respond to your concerns. Considering that the AC meet has started recently, we sincerely hope to have a discussion with you. Please kindly let us know if there is anything else we can address to convince you for upgrading the scores. Thank you very much!
>
> Best wishes,
>
> Authors

---

> ### Author Response · Authors · 2022-12-05
> **Gentle reminder to the reviewer**
>
> Dear Reviewer 5fRb:
>
> Thanks for your constructive reviews. It's been more than **22** days since we submitted our response and we still haven't received your feedback. As the window for discussion is closing, we sincerely hope you can take just a little time to read our response. Please kindly let us know if there is anything else we can address to convince you for upgrading the scores.
>
> Thanks
>
> Authors

---

### Official Review · Reviewer_gnLo · 2022-10-24

**Confidence:** 5
**Correctness:** 2
**Technical Novelty And Significance:** 2
**Empirical Novelty And Significance:** 2
**Recommendation:** 3

**Clarity, Quality, Novelty And Reproducibility:**

This paper is hard to follow. It is not clear what's being addressed and etc. The novelty is very minor and seems like there are a lot of claims that are not supported. Seems to be reproducible.

**Strength And Weaknesses:**

Strength:
- This paper addresses a rather important and interesting problem.
- Many baselines are provided for comparison.
-----------------------------------------------------------
Weakness:
- This paper is poorly written and can be very hard to follow at times.
- This paper seems to be very incremental.
- Seems like the problem is not well-justified. For instance, why is covariate shift different from correlation shift in terms of causal features and environment?
- This paper has many false claims, for example, causal features are NOT invariant across domains.
- While causal features in images are easy to distinguish, how would one define them on graphs? Better examples and justifications are needed.
- Contributions are not justified. Are you addressing the flaws of the existing frameworks? If that is the case, these frameworks are not designed for the specific task defined in this paper. The same goes for the experiments.
- Definition 1 seems not to make sense.

**Summary Of The Paper:**

This paper addresses the problem of covariate shift on graphs. This framework is compared with 14 baselines on both synthetic and real-world data.

**Summary Of The Review:**

This paper introduces a framework that addresses covariate shift. They provide baselines to compare their framework to. However, some of the baselines are not exactly designed for their task. There are some false claims in the paper that should be justified.

---

> ### Author Response · Authors · 2022-11-12
> **Response to Reviewer gnLo (Part 3/3)**
>
> > **[Con4: How would one define causal features on graphs? Better examples and justifications are needed.]**
>
> Please check the introduction in our paper. **In our original paper, we have provided practical examples in our introduction.**  Causal features are the substructures of the entire graphs that truly reflect the predictive property of data. **Taking molecular property predictions as an example**, functional groups are causal features that determine the predictive property of molecules, such as -OH determines the molecular water solubility, -NO2 determines the molecular toxicity. While scaffolds (e.g., carbon rings) are irrelevant patterns, which can be seen as the environments. Similar examples can also be found in studies [20,21].
>
> > **[Con5: Contributions are not justified. Are you addressing the flaws of the existing frameworks? If that is the case, these frameworks are not designed for the specific task defined in this paper. The same goes for the experiments.]**
>
> We are sorry that this was a mistake in our writing, and I hope you can forgive us for the summary of the main contribution in our original paper. We have revised them in our new version. We hope that it will not affect your view of our entire hard work, let me explain the misunderstandings.
>
> - **No, we are not trying to address the flaws of the existing frameworks.** We are exploring one specific type of OOD issue in graph learning: covariate shift, which is of great need but largely unexplored. **Most existing frameworks claim to solve the OOD problem, but they ignore the specific distinction of the OOD problem.** Considering that these frameworks also aim at OOD generalization, **these are the ones that are most relevant to us, so we have to compare and analyze them.** Recent studies [22-24]  have pointed out that these two types of OOD problems (i.e. covariate shift and correlation shift) are different, so it is necessary to clearly distinguish between the them and design solutions for them specifically. Therefore, we choose a more challenging one (covariate shift), analyze this specific OOD problem in detail and design a novel framework to solve it.
>
> - **We respectfully disagree with you that we are solving a "specific task".**  We have repeatedly emphasized in our original paper that these two distribution shifts belong to OOD issues, which are objective problems rather than some specific tasks.
>
> Finally, thank you again for the valuable feedback! **If all your concerns have been resolved, we sincerely hope that you can improve our score. Your support is very important to us, thank you very much!**  Please let us know if you have any other comments/questions, we are more than happy to address them further!
>
>
>
> **References:**
>
> [1] Unsupervised data augmentation for consistency training NeurIPS 2020
>
> [2] Data augmentation with manifold exploring geometric transformations for increased performance and robustness
>
> [3] Low-shot visual recognition by shrinking and hallucinating features, ICCV 2017
>
> [4] Contextual augmentation: data augmentation by words with paradigmatic relations, NAACL 2018
>
> [5] Domain generalization using a mixture of multiple latent domains, AAAI 2020
>
> [6] Deep coral: correlation alignment for deep domain adaptation, ECCV 2016
>
> [7] Model patching: closing the subgroup performance gap with data augmentation, ICLR 2021
>
> [8] Deep domain generalization via conditional invariant adversarial networks, ECCV 2018
>
> [9] Conditional adversarial domain adaptation, NeurIPS 2018
>
> [10] Domain adaptation with conditional distribution matching and generalized label shift, NeurIPS 2020
>
> [11] Just train twice: improving group robustness without training group information, ICML 2021
>
> [12] Invariant models for causal transfer learning, JMLR 2018
>
> [13] On calibration and out-of-domain generalization, NeurIPS 2021
>
> [14] Causal inference by using invariant prediction: identification and confidence intervals
>
> [15] Invariant risk minimization
>
> [16] Invariant risk minimization games, ICML 2020
>
> [17] Out-of-distribution generalization via risk extrapolation (rex), ICML 2021
>
> [18] Invariant Rationalization
>
> [19] Handling Distribution Shifts on Graphs: An Invariance Perspective, ICLR 2022
>
> [20] Discovering Invariant Rationales for Graph Neural Networks, ICLR 2022
>
> [21] Let invariant Rationale Discovery inspire Graph Contrastive Learning, ICML 2022
>
> [22] A Fine-Grained Analysis on Distribution Shift, ICLR 2022
>
> [23] OoD-Bench: Quantifying and Understanding Two Dimensions of Out-of-Distribution Generalization, CVPR 2022
>
> [24] GOOD: A Graph Out-of-Distribution Benchmark, NeurIPS 2022

---

> ### Author Response · Authors · 2022-11-12
> **Response to Reviewer gnLo (Part 2/3)**
>
> > **[Con2: Problem is not well-justified, why is covariate shift different from correlation shift in terms of causal features and environment?]**
>
> Thanks. We have explained the key differences between these two distribution shifts in Abstract, the second paragraph of the Introduction, and Section 2.2, and explicitly gave the mathematical definition of covariate shift. **Both distribution shifts are caused by the environmental features rather than causal features.** Correlation shift denotes that environments and labels establish inconsistent statistical correlations in training and test data. It describes the conditional distribution $P_{tr}(Y|G) \neq P_{te}(Y|G)$. In contrast, covariate shift means that the environmental features in test data are unseen in training data, i.e., $P_{tr}(G) \neq P_{te}(G)$. Since these two distribution shift issues are the conclusions of recent studies [22-24], they are not the main contributions of our paper. Furthermore, due to the page limitations, so we do not give a more detailed discussion of these two distribution shifts in our original paper. Sorry for missing these discussions. **In our revised paper,  we accept your advice and provide a more detailed discussion and comparison in Appendix C.**
>
> > **[Con3: False claims, for example, causal features are NOT invariant across domains & Definition 1 seems not to make sense.]**
>
> Sorry, maybe our description of causal invariance misunderstood you. Below we explain the causal invariance and our assumptions.
>
> - **Following the commonly used assumption [15-20]: causal relationship is invariant across domains.** As the first sentence in Section 2.2 of our paper says, "the labeling rule usually depends on the causal features". Hence,  we assume that **the invariance of causal feature refers to the invariant relationship between causal feature and label, and this relationship is invariant across domains.**
> - **Causal invariance assumption.** In recent years, OOD generalization based on causal invariance assumption has become a research line. There exist a large number of papers [1-22] that are explicitly or implicitly based on the causal invariance. Now **we provide an extensive summary of studies based on causal invariance in recent years, as shown in following table.** The existence of causal and environmental features is their assumptions. **Therefore, in Definition 1 (Assumption 1 in revised paper), we follow them to define our environmental features and causal features.** We also copy the key conclusion from study [22] to support our definition: *"The existence of such invariant features makes OOD generalization possible."*
>
> | $~~~~~~~~~~~~~~~~~~~~$ Approach                           | $~~~~~~~~~~~~~~~~~~~~~~~~~~~~~~~~~~$ Notes                                                        | Invariance Principle                     |
> | ---------------------------------- | ------------------------------------------------------------ | ---------------------------------------- |
> | Data augmentation invariance [1-4] | Representation $\phi(X)$ is invariant to transformation $T(\cdot)$ | $\phi(X)=\phi(T(X))$                     |
> | Distributional invariance [5-7]    | Marginal invariance in environments $e$ and $e'$             | $P^e(\phi(X))=P^{e'}(\phi(X))$           |
> | Distributional invariance [8-11]   | Conditional invariance in environments $e$ and $e'$          | $P^e(\phi(X)\|Y)=P^{e'}(\phi(X)\|Y)$    |
> | Distributional invariance [12-14]  | Sufficiency in environments $e$ and $e'$                     | $P^e(Y\|\phi(X))=P^{e'}(Y\|\phi(X))$       |
> | Risk minimizer invariance [15-18]  | Minimizes risk in all domains                                | Domain-independent predictor $w^*$       |
> | Rationale Invariance [19-21]       | Sufficiency/Independence, Oracle rationale $C$, $S=X\backslash C$ | $f(X)=f(C)$, $Y ⊥S\|C$                    |
> | Ours                               | Sufficiency/Independence, $X_{cau}=T(X)$, $X_{env}=X\backslash X_{cau}$ | $P(Y\|X)=P(Y\|T(X))$, $Y ⊥X_{env}\|X_{cau}$ |

---

> ### Author Response · Authors · 2022-11-12
> **Response to Reviewer gnLo (Part 1/3)**
>
> First of all, thanks for your carefully reading and valuable reviews!  We summarize your **5 concerns** and provide point-to-point responses. **We believe you will change your negative attitude and improve our score if you read them carefully!**
>
> > **[Con1: This paper is poorly written & very incremental & very hard to follow at times]**
>
> Thanks for your time.  According to your comments, our paper is *"poorly written, very incremental, and very hard to follow"*, but could you please tell us **which part is poorly written or very hard to follow?** In addition, we find a complete conflict between you and Reviewer Jykb. We directly copy the comments from Reviewer Jykb: ***“The paper is well-structured, well-written and easy to follow.”***  So we guess it may be because we misunderstood you in our definitions or problem descriptions. **Hence, we carefully check your questions and confusions, and thoroughly revise our paper.** Our changes are as follows:
>
> - **Introduction.** We have emphasized the definitions of causal and environmental features, and add footnotes to make it easier for readers to refer to the definition later in Section 2.2. We also add footnotes for these two distribution shifts, and put detailed comparisons of these two distribution shifts in Appendix C. For our main contribution statement,  we emphasize that we are exploring a new problem: covariate shift, instead of arguing the flaws of existing methods and addressing them.
> - **Section 2.2.** We assume that **the invariance of causal feature refers to the invariant relationship between causal feature and label.** The existence of causal and environmental features is our assumption, so we modify the original Definition 1 to Assumption 1. We have also illustrated the sufficiency conditions and independence conditions mentioned in Assumption 1.
> - **Appendix.** Appendix C adds detailed discussions and comparisons for these two distribution shift issues; Appendix D.2 adds new experimental results on commonly used datasets; Appendix D3 and Figure 6 add more visualizations compared to baseline methods; A comprehensive comparison with the EERM method has been added in Appendix F and Table 7.

---

> ### Author Response · Authors · 2022-12-02
> **Sincerely expecting further discussions with you!**
>
> Dear Reviewer gnLo:
>
> We know that you may be very busy, and we are sorry to disturb you. But we sincerely hope you can take just a little time to read our response. This is very important to us! Please kindly let us know if there is anything else we can address to convince you for upgrading the scores. Thank you very very much!
>
> Best wishes,
>
> Authors

---

> ### Author Response · Authors · 2022-12-05
> **Gentle reminder to the reviewer**
>
> Dear Reviewer gnLo:
>
> Thanks for your efforts in reviewing our paper and providing valuable comments. As the window for discussion is closing, we sincerely hope you can take just a little time to read our response. Please kindly let us know if there is anything else we can address to convince you for upgrading the scores.
>
> Thanks
>
> Authors

---

### Official Review · Reviewer_Jykb · 2022-11-03

**Confidence:** 3
**Clarity, Quality, Novelty And Reproducibility:** The paper is well-structured, well-wr…
**Correctness:** 4
**Technical Novelty And Significance:** 3
**Empirical Novelty And Significance:** 3
**Recommendation:** 6

**Strength And Weaknesses:**

STRENGTHS

This paper tackles an important problem of robust learning of graph classifiers.

Starting with the definitions in Section 2, the paper is easy to follow.

The experiments are extensive and compare the approach to a number of baselines.


WEAKNESSES

Being unfamiliar with graph classifier learning, I found it difficult to develop an intuition as to what "environmental" and "causal" features are when reading the introduction and Figure 1. It only became clear when definitions were provided.

Having said that, I still think it is a bit of a misnomer to call the features "causal", it appears to me that they are simply the relevant ones when it comes to label prediction, the "environmental" ones being simply conditionally independent (and thus irrelevant for classification).


**Summary Of The Paper:**

This paper introduces a method of dealing with covariate shift in graph learning. To this end, the authors define "causal" and "environmental" features (i.e. features that influence label prediction and ones that are conditionally independent of the labels given the former). Subsequently, desiderata of "environmental diversity" and "causal invariance" are introduced and an adversarial learning scheme is devised that leads to learning graphs that keep their "causal features" which should translate to superior out-of-distribution generalization. This is then confirmed in a series of experiments on benchmark data that compare the proposed approach to competing methods.


**Summary Of The Review:**

This paper describes a new method of graph classifier learning by adversarially generating distributions that preserve features that have predictive power towards the labels. The authors provide extensive experiments.

---

> ### Author Response · Authors · 2022-11-12
> **Response to Reviewer Jykb (Part 2/2)**
>
> > **[Con 2: Misnomer to call the features "causal" and "environmental"]**
>
> Thanks, we rudely guess that Reviewer Jykb seems to be unfamiliar with the field of invariant learning and OOD generalization. **These two words are frequently used words in a large amount of literature.** Firstly, we explain these two concepts for you, then we will correct your misunderstandings of these two concepts. Finally, we will list numerous examples to demonstrate that these are two commonly used terms.
>
> - **Causal & Environmental features.** From a data generation view, causal features are substructure of the data, which often reflect the intrinsic and predictive property of the data. **Following the commonly used assumption [1-5]: the relationship between causal features and labels are invariant across different distributions or domains.** Taking image classification as an example, given an image of a dog sitting on the grass, the dog are causal features, which establish a direct causal relationship with the label "dog". The same applies to the case of graph classification, for example, the functional group of the molecule usually determines the chemical properties of the molecule, such as -OH determines the molecular water solubility, -NO2 determines the molecular toxicity.  The causal invariant assumption is the basis for achieving OOD generalization, which has been proofed by many well-known studies [1-5].  In contrast, environmental features are noncausal, which are often irrelevant to the intrinsic property of the data. Hence, they are usually variable and unstable in data, which often lead to distribution shifts.
> - **Your misunderstandings.** According to your comment *"... they (causal features) are simply the relevant ones when it comes to label prediction, ..."*,  you seem to have misunderstood our definition of causal features.  **Correlation significantly differs from the causal relationship in the language of causal inference [4,6,11,12].** Our causal features are defined based on causal relationship, not correlation. Take cow and camel image classification as an example, most of the cows are in the grass and most of the camels are in the desert. At this time, the grass feature and the cow label forms a spurious correlation. But it is clear that the causal features of cow images are cow features, not grass features. If the model's predictions are based on noncausal features, then the model has learned spurious correlations, and is therefore vulnerable to distributional shifts. If the model's label predictions are based on causal features, then the model has learned the causal relationship, which will be robust to distribution shift.
> - **These two words are commonly used terms.** "Causal features" and "environmental features" have gradually become standard academic terms. They have been frequently used in a wide range of work [1-10]. For example, the terms equivalent to "causal feature" [1,5,6] include "rationale" [2,4,7], "causal factor" [8], "causal substructure" [9,10], "causal  elements" [3], "invariant feature [8]". Terms equivalent to "environmental features" [7,9] include "shortcut feature" [4,6], "spurious feature [5]", "spurious substructure" [9], "bias substructure" [10]. Please check these papers and you can easily find these commonly used words. **It can be seen that the OOD generalization community has basically reached a consensus on the understanding of "causal features and environmental features". Hence, following them, we also use these two words in our paper.**
>
> Thank you again for the valuable feedback!  **Finally, if all your concerns have been resolved, we sincerely hope that you can further improve our score. Your support is very important to us, thank you very much!** Please let us know if you have any other comments/questions, we are more than happy to address them further!
>
> [1] Invariant Risk Minimization
>
> [2] Invariant Rationalization, ICML 2020
>
> [3] Out-of-Distribution Generalization via Risk Extrapolation, ICML 2021
>
> [4] Discovering Invariant Rationales for Graph Neural Networks, ICLR 2022
>
> [5] Handling Distribution Shifts on Graphs: An Invariance Perspective, ICLR 2022
>
> [6] Causal Attention for Interpretable and Generalizable Graph Classification, KDD 2022
>
> [7] Graph Rationalization with Environment-based Augmentations, KDD 2022
>
> [8] Learning Causally Invariant Representations for Out-of-Distribution Generalization on Graphs, NeurIPS 2022
>
> [9] Learning Substructure Invariance for Out-of-Distribution Molecular Representations, NeurIPS 2022
>
> [10] Debiasing Graph Neural Networks via Learning Disentangled Causal Substructure, NeurIPS 2022
>
> [11] Interpretation and identification of causal mediation
>
> [12] Models, reasoning and inference

---

> ### Author Response · Authors · 2022-11-12
> **Response to Reviewer Jykb (Part 1/2)**
>
> Thank you for rating our paper as "well-structured, well-written and easy to follow". According to your two concerns, we provide the following responses. **If you are satisfied, we hope you can improve our score**, many thanks!
>
> > **[Con 1: Confused about "environmental" and "causal" features from introduction and Figure 1]**
>
> In our original paper, **we have give detailed definitions and examples of causal and environmental features in the introduction**, such as:
>
> - **In the second paragraph of Introduction**, we have defined the causal and environmental feature. *"... causal features are the substructures of the entire graphs that truly reflect the predictive property of data, while their complementary parts are the environmental features that are noncausal to the predictions..."*
> - **At the beginning of the second page of our paper**, we also give two examples to explain the causal and environmental features. *"... environmental features ladder and tree are different in training and test data" and "functional groups (e.g., NO2) are causal features that determine the predictive property of molecules. While scaffolds (e.g., carbon rings) are irrelevant patterns, which can be seen as the environments..."*
>
> It may be that we do not highlight the causal feature (house) and environmental feature (ladder) in Figure 1. And we apologize for this. Hence, we have made modifications in our revised paper in Figure 1. In addition, **in our revised paper, we also highlight and strengthen the definitions of causal and environmental features and add footnotes in the introduction to make them clear.**

---

> ### Author Response · Authors · 2022-12-05
> **Gentle reminder to the reviewer**
>
> Dear Reviewer Jykb,
>
> Thanks again for your efforts in reviewing our paper. As the discussion deadline is closing, we'd be grateful if you can confirm whether our responses have addressed your concerns. We would be glad to have more discussions if you have further questions and comments.
>
> Thanks.
>
> Authors.

---

### Author Response · Authors · 2022-11-12
**General Responses**

First of all, we sincerely appreciate all reviewers’ time and efforts in reviewing our paper, and thank all reviewers for providing many insightful and constructive suggestions.  We are happy to hear that **we are exploring an important and interesting problem** (Reviewer Jykb, gnLo, 5fRb, rgVp), our paper is **"well-structured, well-written and easy to follow"** (Reviewer Jykb), and **good novelty** (Reviewer rgVp). Here is a summary of our updates:

1. **Clarification**: We clarify our assumption of causal invariance (gnLo), definitions and examples for causal/environmental features (Jykb, gnLo), differences with EERM (5fRb), and some minor issues in descriptions and figures (Jykb, gnLo, 5fRb, rgVp).
2. **Experiments**: We provide more experiments on commonly used datasets (rgVp), and reproduce EERM on graph-level tasks (5fRb).
3. **Comparisons**: We provide **an extensive summary** of existing studies that are based on causal invariance assumption and **detailed discussions** between correlation shift and covariate shift (gnLo). We also provide **detail comparisons with EERM (5fRb) in the following table. This table is also confirmed by the author of EERM.**
4. **Paper revisions**: We put the above major modifications into our paper and highlight the updates in the revision.

|    $~~~~~~~~~~~~~$   Type  |        $~~~~~~~~~~~~~~~~~~~~~~~$     Comparisons  |         $~~~~~~~~~~~~~~~~~$     EERM    |      $~~~~~~~~~~~~~~$     Our AdvCA   |
| :---------------------: | :--------------------------------------------------: | :-------------------------------------: | :----------------------------------------------------: |
|          Scope          |   Is it specifically designed for covariate shift?   |   No     |                          Yes                           |
|      Separability       |   Can environmental/causal features be separated?    |    No     |                          Yes                           |
| Environmental Diversity | Can environmental features be identified explicitly? |                   No                    |                          Yes                           |
| Environmental Diversity | How to model environmental features? |  -   |            Mask model $T_{\theta_1}(\cdot)$            |
| Environmental Diversity | Metric for environmental diversity |  -  |             ${\rm GCS}(\widetilde{P}, P)$              |
| Environmental Diversity | Generation principle for environmental features    | "Blindly" maximize $\mathbb{V}_e[R(e)]$ |         Maximize ${\rm GCS}(\widetilde{P}, P)$         |
|    Causal Invariance    |    Can causal features be identified explicitly?     |      No    |      Yes   |
|    Causal Invariance    |            How to model causal features?             |     -     |            Mask model $T_{\theta_2}(\cdot)$            |
|    Causal Invariance    |       Learning principles for causal features        |  ${\rm min}_\theta\mathbb{V}_e[R(e)]$   |                Sufficiency/Independence                |
|     Generalization      |   Theoretical basis                   |                   IRM                   |                          DRO                           |
|     Generalization      |            Generalization scope            |            $K$ environments   | Robust radius $\rho$:  $D(\widetilde{P}, P) \leq \rho$ |

Finally. we hope our responses can clarify all your confusion and alleviate all concerns. We thank all reviewers' time again. Looking forward to your reply!

---

> ### Author Response · Authors · 2022-11-17
> **Looking forward to your reply!**
>
> Firstly, we thank all the reviewers' time for the review. We have already provided detailed responses. Since the discussion period will end soon, we really hope to have a further discussion with reviewers to see if our responses solve your concerns. Your suggestions and comments are valuable to us.

---

### Author Response · Authors · 2022-11-27
**Dear Area Chair**

We thank all reviewers for their thoughtful and constructive reviews, and we have added related content and experiments in the revised manuscript. Meanwhile, we have provided thorough responses to each reviewer and hope you can look through them. We sincerely expect AC to prompt reviewers to actively participate in the discussion to help us make the paper stronger! We are also open to further discussion if the concerns have not been fully addressed.

---

### Author Response · Authors · 2022-12-05
**Ask for your help!**

Dear Area Chairs,

We want to draw your attention to our submission.
It's been more than **22** days since we submitted our response but **we still haven't received ANY feedback!**
After carefully considering the reviewers' comments, we add a large number of experiments and also provide point-to-point answers to address all the concerns of reviewers. Despite the substantial time and effort that we spend on answering the reviewers' questions, there is sadly no discussion for our submission.

The author-reviewer discussion deadline is approaching. We sincerely hope that our efforts and improvements will be taken into consideration. We would appreciate it if you could help us to remind the reviewers.

Thanks for your attention and participation.

Authors

---

### Decision · Program_Chairs · 2023-01-20

**Decision:**

Reject

**Justification For Why Not Higher Score:**

unclear motivation of using causality

the assumptions under which the data augmentation method could work is not clear.

**Justification For Why Not Lower Score:**

n/a

**Metareview: Summary, Strengths And Weaknesses:**

This paper studies OOD generalization on graphs by considering covariate shift, i.e., the distribution of graph inputs are different in new domains. To address this problem, the authors propose an adversarial causal augmentation method to explore the diverse distribution of environments.

While the proposed problem is interesting and the proposed method is interesting, there are several concerns on the technical quality and novelty. First, the authors overuse the term “causality” in the paper. By this paper’s definition, the causal features are the stable features while the environmental features are changing features. It is not necessary to define a causal model as the proposed method does not make use of any property of the causal generative process, e.g., modularity or interventional causal effects. Second, the authors claim that the two set of features can be separated, however, there is no theoretical evidence or empirical results to demonstrate it. Third, it is unclear why the proposed data augmentation strategy could cover possible structures in the new test domain. The assumption under which the data augmentation could work is not provided. Given these concerns, I would recommend rejection of this paper in its current form.  Although we think the paper is not ready for ICLR in this round, we believe that the paper would be a good one if the concerns can be well addressed.